bioinformatics/plant science/microbiology

*Xanthomonas translucens*, Zn-binding proteins, *in silico*, Zn homeostasis, plant–pathogen interactions

**Author for correspondence:**
Shailender Kumar Verma
e-mail: skverma@cuhimachal.ac.in

†Present address: Department of Biological Chemistry, John Innes Centre, Norwich NR4 7UH, UK.

# Zinc binding proteome of a phytopathogen *Xanthomonas translucens* pv. *undulosa*

Ankita Sharma, Dixit Sharma and

Shailender Kumar Verma†

Centre for Computational Biology and Bioinformatics, School of Life Sciences, Central University of Himachal Pradesh, Kangra, Himachal Pradesh 176206, India

AS, 0000-0003-1360-5810; DS, 0000-0002-9911-9364;
SKV, 0000-0002-4192-933X

*Xanthomonas translucens* pv. *undulosa* (*Xtu*) is a proteobacteria which causes bacterial leaf streak (BLS) or bacterial chaff disease in wheat and barley. The constant competition for zinc (Zn) metal nutrients contributes significantly in plant–pathogen interactions. In this study, we have employed a systematic *in silico* approach to study the Zn-binding proteins of *Xtu*. From the whole proteome of *Xtu*, we have identified approximately 7.9% of proteins having Zn-binding sequence and structural motifs. Further, 115 proteins were found homologous to plant–pathogen interaction database. Among these 115 proteins, 11 were predicted as putative secretory proteins. The functional diversity in Zn-binding proteins was revealed by functional domain, gene ontology and subcellular localization analysis. The roles of Zn-binding proteins were found to be varied in the range from metabolism, proteolysis, protein biosynthesis, transport, cell signalling, protein folding, transcription regulation, DNA repair, response to oxidative stress, RNA processing, antimicrobial resistance, DNA replication and DNA integration. This study provides preliminary information on putative Zn-binding proteins of *Xtu* which may further help in designing new metal-based antimicrobial agents for controlling BLS and bacterial chaff infections on staple crops.

## 1. Introduction

The bacterial diseases of the crop plants place a major restraint on crop production and result in significant global food production losses and food security [1,2]. *Xanthomonas translucens* pv. *undulosa* (*Xtu*) is a gram-negative bacterial pathogen of crop plants, wheat and barley [3]. The infection of *Xtu* on wheat and

barley causes bacterial leaf streak (BLS) and bacterial chaff diseases. BLS disease is distributed worldwide and brings about 30–40% yield losses [4]. Wheat crop is one of the staple food crops which feed approximately 30% of the total population and act as a rich source of macro- and micro-nutrients. The nutrients acquisition by the pathogen and in response the nutrients immunity provided by the plant host play critical roles in plant–pathogen interactions [5]. Also, the bacterial pathogens have a complex association between the metabolic processes, regulation of expression and functioning of virulence factors [6–8]. The efficient utilization of the available nutrients is required by the pathogenic bacteria to survive inside the host milieu. The constant competition for the trace transition metal ions is one of the key factors at the traffic circle of nutrient metabolism and virulence [9,10]. The transition metal ions are necessary for the survival of all the living organisms. Approximately one-third of all proteins contain one or more metal ions as cofactor for their structural stability and functional activity [11].

The transition metal, zinc (Zn) represents the second most abundant metal cofactor after iron [10]. Zinc is stable as divalent cation ($Zn^{2+}$) with complete filled outer shell $d$-orbital ($3d^{10}$) and has no redox activity [12]. Zinc metal performs a variety of catalytic, structural and regulatory activities in a number of proteins [13]. Further, these activities assist in various biological and cellular processes like gene expression, biosynthesis of extracellular peptidoglycans and amino acids, reactive oxygen species (ROS) detoxifications, DNA repair, production of virulence-related proteins and antibiotic resistance [13–15]. Zinc ion is the highly competitive divalent metal ion of Irving–Williams series after copper (Cu) and therefore can easily replace the other metals from their cognate metalloenzyme [16]. The intracellular pool of 'free' Zn in the cells must be kept low because of the high chelating ability of Zn [17]. In bacteria, the Zn concentration may vary from the range of 0.1–1.0 mM [18]. Although zinc is an essential micronutrient, its higher concentrations result in significant toxicity to the bacterial cell [18]. Therefore, a great challenge for the bacterial pathogens is to procure an adequate concentration of Zn for maintaining their growth and survival during the infection. Previously, it is a known fact that intracellular Zn-binding, Zn-sensing and import or export of Zn ions helps in maintaining Zn homeostasis in bacteria [19–21]. Studies on Zn-homeostatic mechanisms regulation of most of the bacteria like *E. coli*, *B. subtilus*, *B. anthracis*, *Staphylococcus*, *Streptococcus* has been made earlier [22–27], which stated that Zn uptake and efflux system play roles in bacterial virulence. Earlier, a systematic study on zinc proteome of *E. coli* was made using the assay of radioactive $Zn^{2+}$ binding on the total proteins fractionated by two-dimensional gel electrophoresis [28]. The report stated that most of the newly identified Zn-binding proteins do not have known Zn-binding motifs that were earlier identified in higher eukaryotes. Further, the usage of traditional experimental techniques is restricted for the prediction of metalloproteins at complete proteome level. This is due to time resolution, high cost, sensitivity and need of more expertise to prepare sample and to handle specialized equipments precisely [29,30]. Therefore, the call of the hour is to use new high-throughput technologies of post-genomic era for genome-wide identification of metalloproteins [15,31–33].

A systematic computational approach has been used in the current study to identify and characterize the Zn-binding proteins from the whole proteome of the *Xtu* strain 4699 [34]. Furthermore, the identified Zn-binding proteins were also checked for their probable involvement in plant–pathogen interactions and virulence. This primarily study provides us putative Zn-binding proteins which probably act as targets for controlling BLS.

# 2. Material and methods

## 2.1. Proteome extraction and identification of Zn-binding proteins

The whole proteome of *Xtu* was downloaded from Refseq National Centre for biotechnology Information (NCBI) server. The complete proteome has 3536 proteins and all these were examined for the presence of Zn-binding motifs using MetalPDB [35]. MetalPDB is a database of metalloproteins which provide features of metalloproteins and their minimal functional sites. The information of metalloproteins stored in MetalPDB is drawn from PDB, Pfam, CATH and SCOP databases. We have collected the information of Zn-binding proteins from MetalPDB and prepared a local dataset of Zn-binding proteins to perform stand-alone blastp search on the whole proteome of *Xtu* at expect value (e-value) 0.00001. The proteins which were found homologous to the Zn dataset of MetalPDB at e-value ≤ 0.00001 were further selected.

The short-listed proteins were modelled by protein homology/analogy recognition engine v. 2.0 (Phyre2) program [36]. This is done because three-dimensional structure of the proteins aid to determine its interactions with metal ions. Phyre2 server is built on hidden Markov model for creation of three-dimensional structure of the protein. The high-throughput modelled proteins, having confidence and query coverage more than or equal to 90% and 50%, respectively, were manually chosen. The modelled proteins were scanned for the putative Zn-binding structural motifs using metal ion binding site prediction and docking server (MIB) [37], which is built on a fragment transformation method. In this method, the query protein was aligned to the metal binding templates that were extracted from metal bound proteins present in PDB. The templates represent the local structure of metal binding residues within 3.5 Å. According to MIB server, a metal binding site had to contain a metal ion and at least two residues to quantify as a metal ion binding residues template. Each cluster, after sequence and structural similarity, acquires a particular score, which is used for prediction of metal binding sites. For the evaluation of sequence similarity, MIB server uses BLOSUM62 matrix, and for calculation of structural similarity root mean square deviation of $C_\alpha$ atoms of the alignments was used. At more than 95% specificity threshold, MIB server predicts the Zn-binding sites with 94.8% accuracy and 71.1% sensitivity [37]. Ligplot$^+$ visualization tool was further used to check the interactions of MIB docked $Zn^{2+}$ metal ion with the protein [38]. We found that interacting residues and interaction radii provided by MIB vary in a wide range. Therefore, the interaction distance was raised from primary sphere (up to 3.5 Å) (provided by MIB server) to secondary sphere (5 Å). Also, it was stated earlier that second shell of interactions helps in stabilizing metal binding site, raises metal affinity and plays a role in determining physical properties of transition metal complexes [39–41]. Further, we have ignored the proteins which only bind to backbone ($C_\alpha$) chain atoms. The proteins which were found to interact with $Zn^{2+}$ ion up to 5 Å were finally selected as putative Zn-binding proteins.

## 2.2. Functional annotation, gene ontology analysis and localization prediction of Zn-binding proteins

The selected Zn-binding proteins were explored for functional domains, family and super-families using different bioinformatics databases: InterProScan [42], Pfam [43] and NCBI-CDD [44]. The broad classification of these proteins was done by literature reviews of identified domains and families. Further, the clustergram was generated using MEGA6 [45], BioEdit [46] and EvolView [47] servers. To construct and visualize the gene ontology (GO) [48] based molecular function and biological process networks, a Cytoscape [49] plug-in ClueGO v. 2.3.3 [50] was used. In these networks, each node indicates particular GO terms (biological or molecular) and edge indicates connections between GO terms based on their gene association. The significance of the particular GO term was indicated by the size of that particular node. The statistical kappa score method [51] was used in order to determine the functional grouping of these identified GO terms. Further, the selected Zn-binding proteins were analysed for their subcellular localization using bioinformatics utilities viz. PSORTb, CELLO and SOSUI-GramN [52–54]. The consensus of these was taken in order to predict precise localization.

## 2.3. Prediction of putative Zn-binding proteins probably involved in plant–pathogen interactions

The predicted Zn-binding proteins were examined for their participation in plant–pathogen interactions using blastp search against the experimentally validated virulent and effector proteins of bacterial plant pathogens present in Pathogen–Host Interaction database (PHI-base) [55]. The proteins that were found homologous at e-value $\leq 0.0001$ were short-listed and probably considered to play roles in plant–pathogen interactions. Further, these short-listed proteins were scanned for their secretory nature using neural network-based computational servers SignalP [56] TatP [57] and SecretomeP [58], respectively. The proteins showing presence of signal peptide, Tat motif or proteins having Sec score more than or equal to 0.5 were selected as probable secretory proteins. To avoid the false positive prediction of secretory Zn-binding proteins, we have checked the presence of transmembrane α-helix in these selected proteins using transmembrane hidden Markov model (TMHMM) [59] and hidden Markov model for topology prediction (HMMTOP) [60] servers. The proteins having single or no transmembrane helix were short-listed further and referred as putative secretory Zn-binding proteins.

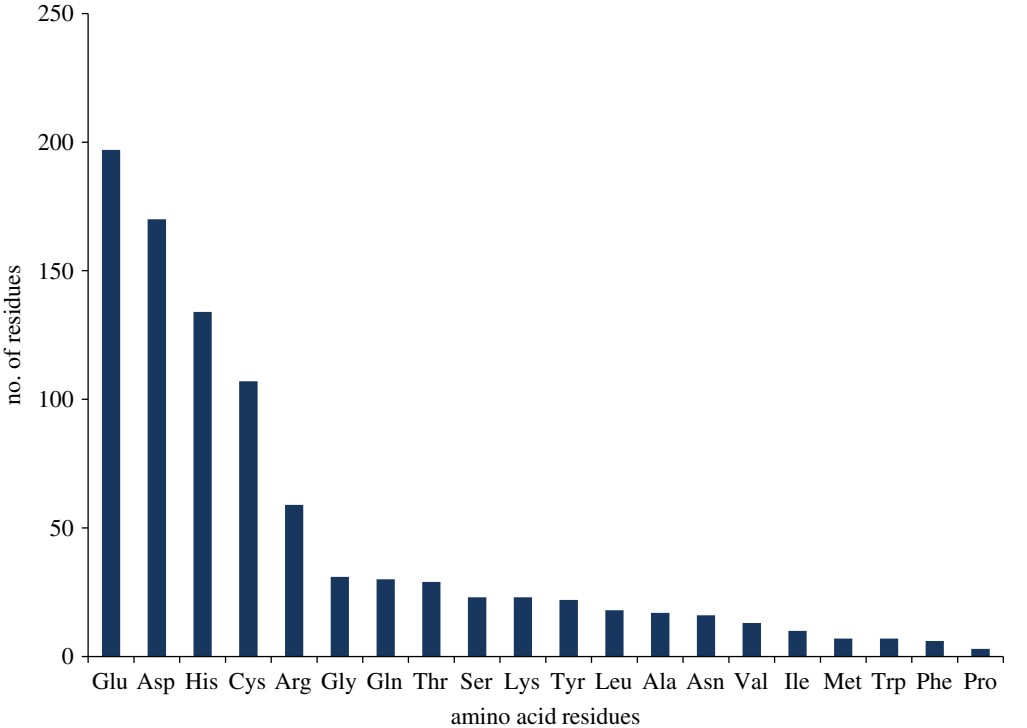

**Figure 1.** Interactions of $Zn^{2+}$ metal ion with amino acid residues. The graph here represents the amino acid residues interacting with $Zn^{2+}$ ion. The X-axis represents here the name of amino acid residues and Y-axis is showing number of amino acid residues. The frequently interacting residues with Zn metal ions were Glu, Asp, His and Cys.

## 3. Results

### 3.1. Zinc-binding proteins and their binding patterns presented by *Xtu* proteome

Out of the complete proteome of *Xtu*, 346 proteins showed the occurrence of putative Zn-binding sequence motifs. Based on the modelling criteria and manual verification of structures' intactness, 335 proteins were short-listed further (electronic supplementary material, table S1). After Zn-protein interactions analysis by Liglpot⁺, 279 proteins showed interactions up to 5 Å which were selected further as putative Zn-binding proteins (electronic supplementary material, table S2). The interacting amino acid residues in Zn-binding proteins were Glu > Asp > His > Cys > Arg > Gly > Gln > Thr > Ser = Lys > Tyr > Leu > Ala > Asn > Val > Ile > Met = Trp > Phe > Pro (figure 1).

### 3.2. Functional annotation and cellular compartmentalization of predicted Zn-binding proteins

The functional domain investigation of the scanned Zn-binding proteins showed the foremost existence of short chain dehydrogenase/reductase (SDR), response regulator receiver, alcohol dehydrogenase/ GroES (ADH/GroES), tRNA synthetase and ABC transporter domains. Based on the literature studies of the identified domains, the Zn-binding proteins were widely classified into 13 classes. Most of the proteins associated with the classes of metabolic process (122), proteolysis (24), protein biosynthesis (22), transport (21), cell signalling (20), protein folding (13), DNA repair (12), transcription regulation (12), response to oxidative stress (11), RNA processing (9), antimicrobial resistance (7), DNA replication (4) and DNA integration (2). The detailed description is displayed in figure 2 and electronic supplementary material, table S3. The examination of cellular compartmentalization of short-listed Zn-binding proteins indicates that 76% of the proteins reside in cytoplasm, 15.8% in periplasm, 7.2% in inner-membrane, followed by outer-membrane (0.7%) and extracellular space (0.3%) (figure 3; electronic supplementary material, table S3).

### 3.3. Gene ontology network analysis of Zn-binding proteins

The GO biological network of Zn-binding proteins was configured on 13 kappa score groups which contain 122 GO biological process terms and 623 GO biological terms connections (figure 4;

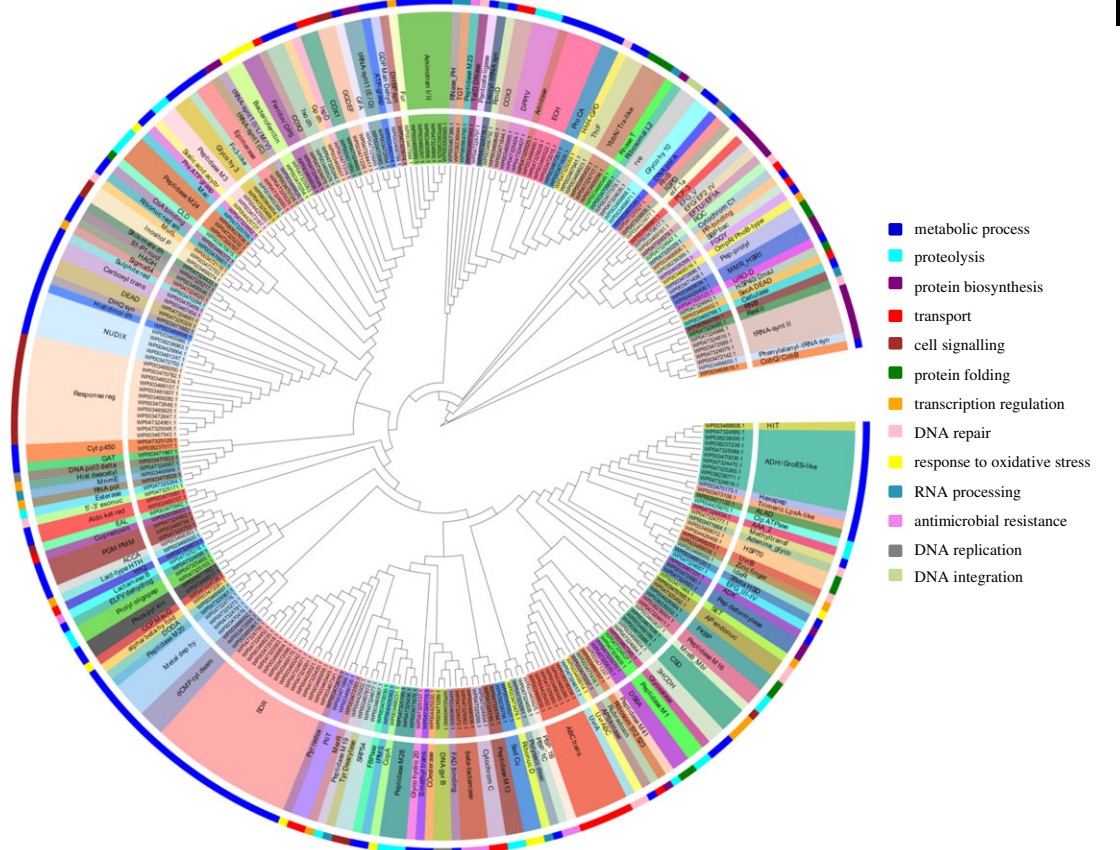

**Figure 2.** The functional classification of Zn-binding proteins. The functional classification of identified Zn-binding proteins was done on the basis of their domains. The clustergram was constructed by MEGA6 [45]. The inner circle of the clustergram represents sequence ID of the proteins. The middle and outer circle represent the functional domain and broad categories of the Zn-binding proteins. The most common domains in Zn-binding proteins of *Xtu* were SDR, response regulator, ADH/GroES, tRNA synthetase and ABC transporter. These proteins have diverse roles in metabolic processes, proteolysis, protein synthesis and transport.

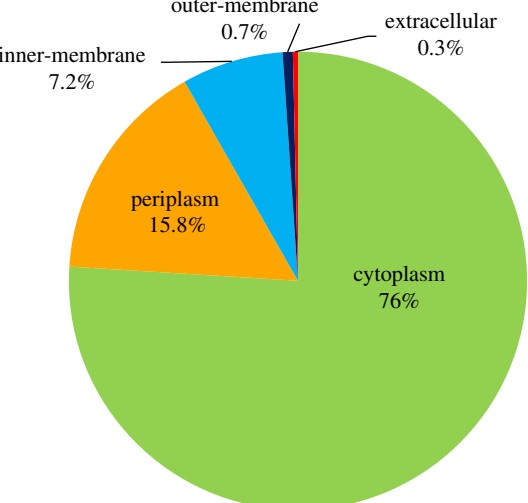

**Figure 3.** Subcellular localization of predicted Zn-binding proteins of *Xtu*. The pie chart shown here represents subcellular localization of predicted Zn-binding proteins of *Xtu*. Most of the proteins found to be localized in cytoplasm. Small number of proteins found in periplasm and inner-membrane. A little fraction of proteins found in outer-membrane and extracellular space.

electronic supplementary material, table S4). The GO terms which showed their presence in more than two functional groups indicate their roles in multiple biological processes. The GO biological terms cellular iron ion homeostasis and lipid modification represented the most significant groups in the

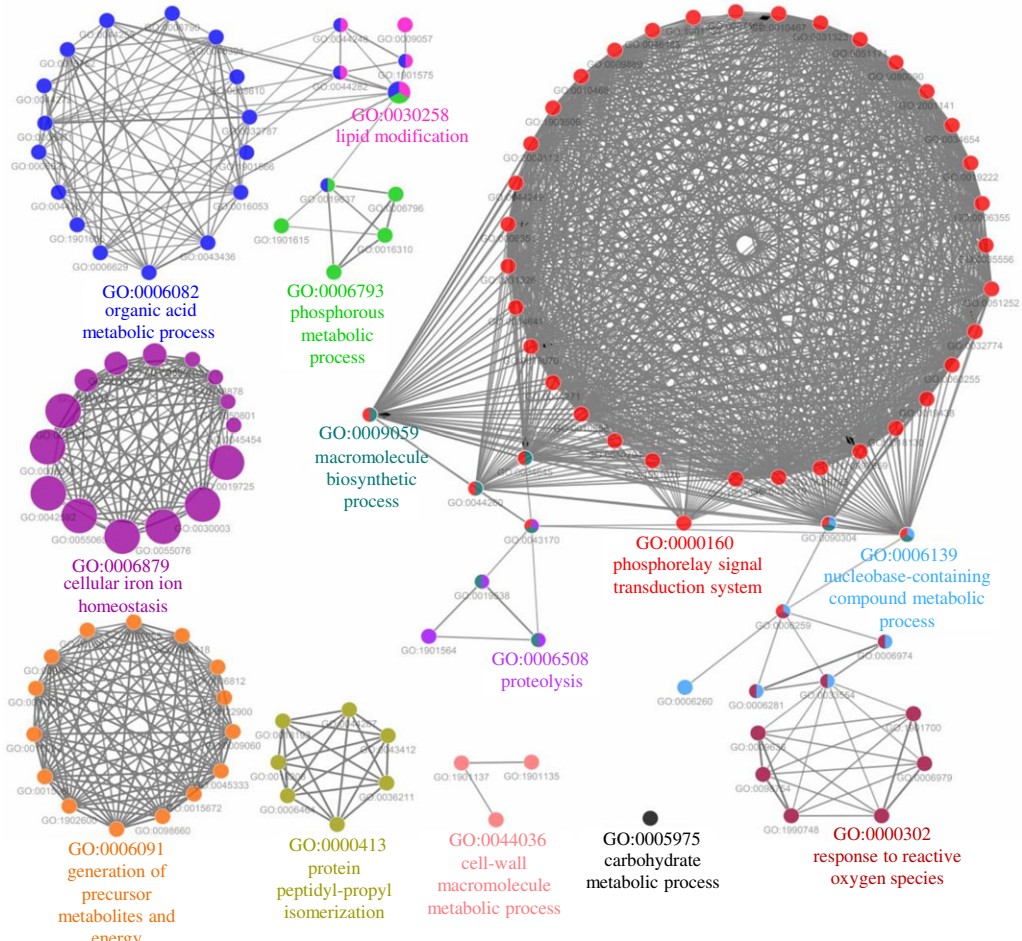

**Figure 4.** GO biological process network of Zn-binding proteins of *Xtu*. The ClueGO biological process network of Zn-binding proteins was constructed at kappa score ≥0.4. The circle here represents the node which indicates particular GO biological term. The colour of the node represents the particular GO group and mixed colour of the node indicates that node belongs to multiple groups. A total of 13 kappa score groups were found in this network, out of which cellular iron ion homeostasis and lipid modification GO biological processes were most significant.

GO biological process network. In order to check the involvement of Zn-binding proteins in biological processes, the analysis of the network was made and the number of interactions was estimated. The cellular macromolecule metabolic process (GO:0044260) and organic substance biosynthetic process (GO:1901576) were found as the most connected nodes with 69 links of each (electronic supplementary material, table S4). Further, to determine the molecular activities of scrutinized Zn-binding proteins the GO molecular function network was built on 18 kappa score groups. The network has 71 GO molecular function terms (nodes) and 136 connections (figure 5; electronic supplementary material, table S5). The most significant groups in the network were transition metal ion binding, iron ion binding, metallopeptidase activity and oxidoreductase acting on the CH–OH group of the donors and NAD or NADP as acceptor. The GO:0046872 and GO:0043169 were the most coupled GO molecular terms with 48 and 46 links of each, respectively (electronic supplementary material, table S5).

## 3.4. Zinc-binding proteins probably involved in plant–pathogen interactions

The Zn-binding proteins contribute in plant–pathogen interactions. Therefore, the selected 279 Zn-binding proteins were examined for their probable role in plant–pathogen interactions. Out of 279 proteins, 115 proteins were found homologous to PHI-base (electronic supplementary material, table S6). Among 115 proteins, 11 proteins were found to be secretory with single or no transmembrane

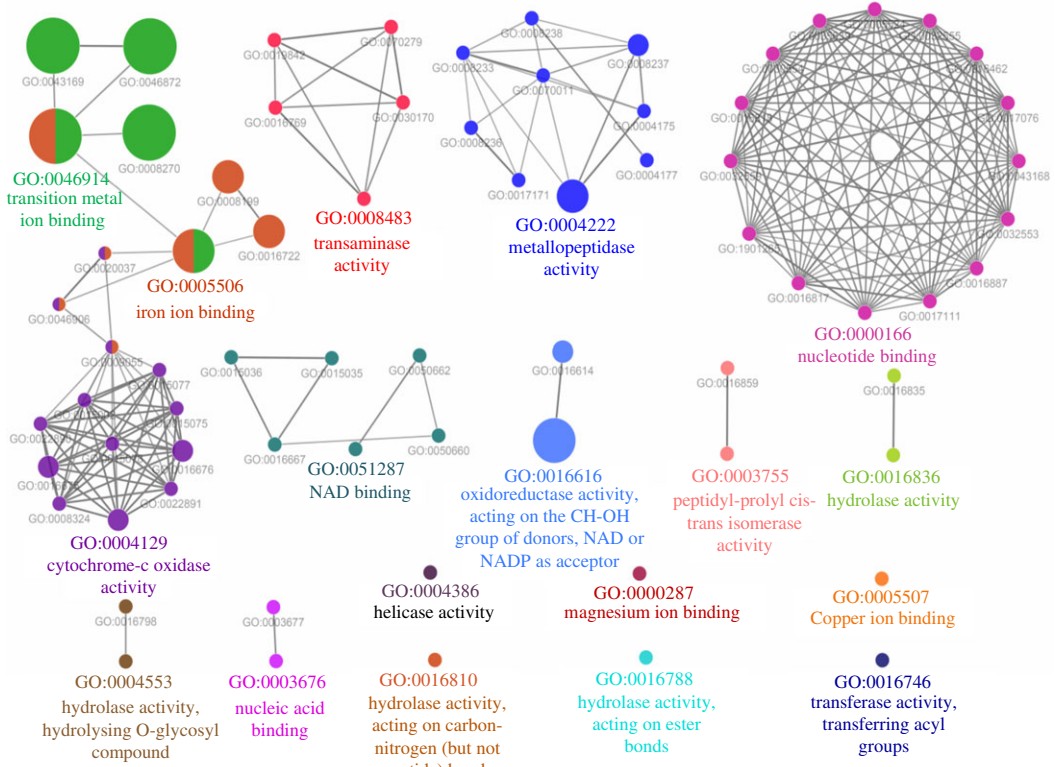

**Figure 5.** GO Molecular functional network of Zn-binding proteins of *Xtu*. The ClueGO molecular function network of Zn-binding proteins was built on kappa score ≥0.4. The circle here indicates the node with particular GO molecular term. The node colour represents the group to which they belong. The mixed coloured node indicates their presence in multiple molecular functions. A total of 18 kappa score groups were found in this network. The most significant groups were transition metal ion binding, iron ion binding, metallopeptidase activity and oxidoreductase activity.

helix (electronic supplementary material, table S7). These identified homologous Zn-binding proteins are probably considered to play important roles in bacterial virulence, survival and plant–pathogen interactions. The details of functional domains and categories of these proteins are given in table 1 and figure 6.

# 4. Discussion

The present study focuses on the bioinformatic identification of potential Zn-binding proteins within the whole proteome of *Xtu* and their putative roles in its physiology and virulence. We have found 279 putative Zn-binding proteins which represent 7.9% of the whole proteome of *Xtu*. Previously, it was reported that Zn proteome of an organism varies in range from 4% to 10% of the whole proteome [126]. The predicted Zn proteome of *Xtu* not only contain Zn-binding proteins which exist in proper coordination (biological relevant assembly), but also Zn-substituted proteins and proteins participate as Zn buffering system. Earlier, it was stated that cytosolic Zn-binding proteins, transporters localized in cytoplasmic membranes and sensors of cytoplasmic free Zn ions are the molecules involved in Zn-homeostatic mechanisms of a cell [17]. Glutamate, aspartate, histidine and cysteine were found as frequently interacting residues in the binding pocket of Zn-binding proteins. Earlier, it was known that oxygen atoms of glutamate and aspartate, nitrogen atom of histidine and sulfur atom of cysteine were commonly interacting with Zn metal ions [127]. Some predicted proteins have one or two coordinates, i.e. do not have stable coordination. This may be due to the reason that MetalPDB and MIB rely on asymmetric units rather than biologically relevant assemblies [35,37]. Secondly, it may be because some ligand atom (water, inorganic or organic molecule) may bind to metal ion for their stable coordination in the biological active form [12].

**Table 1.** Zn-binding proteins probably involved in pathogen–host interactions.

| s. no. | sequence ID of putative Zn-binding protein | subcellular localization | functional domain/family | broad functional class | reference |
|---|---|---|---|---|---|
| 1 | WP_004426362.1 | cytoplasmic | 2-isopropylmalate synthase, bacterial-type | metabolic process | [61] |
| 2 | WP_003471682.1 | cytoplasmic | 3-dehydroquinate synthase | metabolic process | [62] |
| 3 | WP_003468654.1 | cytoplasmic | 3-hydroxyacyl-CoA dehydrogenase, NAD-binding domain | metabolic process | [63] |
| 4 | WP_047324684.1 | cytoplasmic | 3-hydroxyacyl-CoA dehydrogenase, NAD-binding domain | metabolic process | [63] |
| 5 | WP_003472081.1 | periplasmic | alpha/beta hydrolase fold | metabolic process | [64] |
| 6 | WP_003465615.1 | cytoplasmic | CobQ/CobB/MinD/ParA nucleotide-binding domain | metabolic process | [65] |
| 7 | WP_038237017.1 | cytoplasmic | cytochrome P450 | metabolic process | [66] |
| 8 | WP_047325129.1 | cytoplasmic | cytochrome P450 | metabolic process | [66] |
| 9 | WP_003465522.1 | cytoplasmic | enoyl-CoA hydratase/ isomerase | metabolic process | [67] |
| 10 | WP_003468480.1 | cytoplasmic | enoyl-CoA hydratase/ isomerase | metabolic process | [67] |
| 11 | WP_047325628.1 | cytoplasmic | enoyl-CoA hydratase/ isomerase | metabolic process | [67] |
| 12 | WP_003465369.1 | cytoplasmic | fibronectin type III-like domain | metabolic process | [68] |
| 13 | WP_003466789.1 | cytoplasmic | GDP-mannose 4,6 dehydratase | metabolic process | [69] |
| 14 | WP_003469871.1 | periplasmic | glycoside hydrolase family 10 | metabolic process | [70–72] |
| 15 | WP_003469868.1 | periplasmic | glycoside hydrolase family 10 | metabolic process | [70–72] |
| 16 | WP_047325488.1 | periplasmic | glycoside hydrolase family 3 | metabolic process | [70–72] |
| 17 | WP_080964854.1 | periplasmic | glycoside hydrolase family 3 | metabolic process | [70–72] |
| 18 | WP_047324805.1 | cytoplasmic | imidazoleglycerol-phosphate dehydratase | metabolic process | [73] |

(*Continued.*)

| s. no. | sequence ID of putative Zn-binding protein | subcellular localization | functional domain/family | broad functional class | reference |
|---|---|---|---|---|---|
| 19 | WP_003468529.1 | cytoplasmic | isocitrate/isopropylmalate dehydrogenase | metabolic process | [74] |
| 20 | WP_047325244.1 | cytoplasmic | isocitrate/isopropylmalate dehydrogenase | metabolic process | [74] |
| 21 | WP_047324950.1 | cytoplasmic | NAD-dependent epimerase/ dehydratase family | metabolic process | [75] |
| 22 | WP_047325529.1 | cytoplasmic | NAD-dependent epimerase/ dehydratase family | metabolic process | [75] |
| 23 | WP_047324797.1 | cytoplasmic | pantoate ligase | metabolic process | [76] |
| 24 | WP_003468806.1 | cytoplasmic | phosphopantetheine attachment site | metabolic process | [77] |
| 25 | WP_047325261.1 | inner-membrane | polysaccharide deacetylase | metabolic process | [78] |
| 26 | WP_047324952.1 | cytoplasmic | pre-ATP-grasp domain | metabolic process | [79] |
| 27 | WP_047324777.1 | cytoplasmic | S-adenosyl-L-methionine-dependent methyltransferase | metabolic process | [80] |
| 28 | WP_003465242.1 | cytoplasmic | short-chain dehydrogenase/ reductase SDR | metabolic process | [81,82] |
| 29 | WP_003465445.1 | cytoplasmic | short-chain dehydrogenase/ reductase SDR | metabolic process | [81,82] |
| 30 | WP_003466959.1 | cytoplasmic | short-chain dehydrogenase/ reductase SDR | metabolic process | [81,82] |
| 31 | WP_003467341.1 | cytoplasmic | short-chain dehydrogenase/ reductase SDR | metabolic process | [81,82] |
| 32 | WP_003468804.1 | cytoplasmic | short-chain dehydrogenase/ reductase SDR | metabolic process | [81,82] |
| 33 | WP_003469170.1 | cytoplasmic | short-chain dehydrogenase/ reductase SDR | metabolic process | [81,82] |
| 34 | WP_003472018.1 | cytoplasmic | short-chain dehydrogenase/ reductase SDR | metabolic process | [81,82] |
| 35 | WP_003472098.1 | cytoplasmic | short-chain dehydrogenase/ reductase SDR | metabolic process | [81,82] |
| 36 | WP_004425521.1 | periplasmic | short-chain dehydrogenase/ reductase SDR | metabolic process | [81,82] |
| 37 | WP_047324452.1 | cytoplasmic | short-chain dehydrogenase/ reductase SDR | metabolic process | [81,82] |

(*Continued.*)

| s. no. | sequence ID of putative Zn-binding protein | subcellular localization | functional domain/family | broad functional class | reference |
|---|---|---|---|---|---|
| 38 | WP_047324671.1 | cytoplasmic | short-chain dehydrogenase/ reductase SDR | metabolic process | [81,82] |
| 39 | WP_047324755.1 | cytoplasmic | short-chain dehydrogenase/ reductase SDR | metabolic process | [81,82] |
| 40 | WP_047324951.1 | cytoplasmic | short-chain dehydrogenase/ reductase SDR | metabolic process | [81,82] |
| 41 | WP_047325420.1 | cytoplasmic | short-chain dehydrogenase/ reductase SDR | metabolic process | [81,82] |
| 42 | WP_047325583.1 | cytoplasmic | short-chain dehydrogenase/ reductase SDR | metabolic process | [81,82] |
| 43 | WP_003473108.1 | cytoplasmic | trimeric LpxA-like | metabolic process | [83] |
| 44 | WP_003470036.1 | cytoplasmic | alcohol dehydrogenase GroES-like domain | metabolic process | [84–86] |
| 45 | WP_038237239.1 | cytoplasmic | alcohol dehydrogenase GroES-like domain | metabolic process | [84–86] |
| 46 | WP_038238095.1 | cytoplasmic | alcohol dehydrogenase GroES-like domain | metabolic process | [84–86] |
| 47 | WP_038238771.1 | cytoplasmic | alcohol dehydrogenase GroES-like domain | metabolic process | [84–86] |
| 48 | WP_047324470.1 | cytoplasmic | alcohol dehydrogenase GroES-like domain | metabolic process | [84–86] |
| 49 | WP_047324585.1 | cytoplasmic | alcohol dehydrogenase GroES-like domain | metabolic process | [84–86] |
| 50 | WP_047324619.1 | cytoplasmic | alcohol dehydrogenase GroES-like domain | metabolic process | [84–86] |
| 51 | WP_047325089.1 | cytoplasmic | alcohol dehydrogenase GroES-like domain | metabolic process | [84–86] |
| 52 | WP_047325265.1 | cytoplasmic | alcohol dehydrogenase GroES-like domain | metabolic process | [84–86] |
| 53 | WP_004425530.1 | periplasmic | carboxylesterase family | metabolic process | [87,88] |
| 54 | WP_003470615.1 | cytoplasmic | cyclophilin-type peptidyl- prolyl cis-trans isomerase/CLD | protein folding | [89,90] |
| 55 | WP_004426584.1 | periplasmic | FKBP-type peptidyl-prolyl cis-trans isomerase domain | protein folding | [91–93] |

| s. no. | sequence ID of putative Zn-binding protein | subcellular localization | functional domain/family | broad functional class | reference |
|---|---|---|---|---|---|
| 56 | WP_038239202.1 | periplasmic | FKBP-type peptidyl-prolyl cis-trans isomerase domain | protein folding | [91–93] |
| 57 | WP_047324842.1 | cytoplasmic | HSP40/DnaJ peptide-binding | protein folding | [93,94] |
| 58 | WP_003468012.1 | cytoplasmic | Hsp70 protein/DnaK | protein folding | [93,94] |
| 59 | WP_004426468.1 | cytoplasmic | Hsp70 protein/DnaK | protein folding | [93,94] |
| 60 | WP_003471408.1 | cytoplasmic | Peptidyl-prolyl cis-trans isomerase domain | protein folding | [91–93] |
| 61 | WP_003470942.1 | cytoplasmic | EAL domain | cell signalling | [95] |
| 62 | WP_003471228.1 | cytoplasmic | GGDEF domain | cell signalling | [96] |
| 63 | WP_047324417.1 | cytoplasmic | GGDEF domain | cell signalling | [96] |
| 64 | WP_003465234.1 | cytoplasmic | response regulator receiver domain | cell signalling | [97,98] |
| 65 | WP_003465620.1 | cytoplasmic | response regulator receiver domain | cell signalling | [97,98] |
| 66 | WP_003466157.1 | cytoplasmic | response regulator receiver domain | cell signalling | [97,98] |
| 67 | WP_003467543.1 | cytoplasmic | response regulator receiver domain | cell signalling | [97,98] |
| 68 | WP_003469200.1 | cytoplasmic | response regulator receiver domain | cell signalling | [97,98] |
| 69 | WP_003469280.1 | cytoplasmic | response regulator receiver domain | cell signalling | [97,98] |
| 70 | WP_003470782.1 | cytoplasmic | response regulator receiver domain | cell signalling | [97,98] |
| 71 | WP_003472647.1 | cytoplasmic | response regulator receiver domain | cell signalling | [97,98] |
| 72 | WP_003472648.1 | cytoplasmic | response regulator receiver domain | cell signalling | [97,98] |
| 73 | WP_003481607.1 | cytoplasmic | response regulator receiver domain | cell signalling | [97,98] |
| 74 | WP_047324961.1 | cytoplasmic | response regulator receiver domain | cell signalling | [97,98] |
| 75 | WP_047325048.1 | cytoplasmic | response regulator receiver domain | cell signalling | [97,98] |
| 76 | WP_003466285.1 | inner-membrane | ABC transporter | transport | [18,99] |
| 77 | WP_003466345.1 | inner-membrane | ABC transporter | transport | [18,99] |
| 78 | WP_003468030.1 | inner-membrane | ABC transporter | transport | [18,99] |
| 79 | WP_004425452.1 | inner-membrane | ABC transporter | transport | [18,99] |

(*Continued.*)

| s. no. | sequence ID of putative Zn-binding protein | subcellular localization | functional domain/family | broad functional class | reference |
|--------|--------------------------------------------|--------------------------|--------------------------|------------------------|-----------|
| 80 | WP_038238334.1 | inner-membrane | ABC transporter | transport | [18,99] |
| 81 | WP_047325000.1 | inner-membrane | ABC transporter | transport | [18,99] |
| 82 | WP_003469645.1 | inner-membrane | cytochrome c oxidase subunit III | transport | [100,101] |
| 83 | WP_003471844.1 | inner-membrane | cytochrome c oxidase subunit III | transport | [100,101] |
| 84 | WP_003466735.1 | cytoplasmic | pilus retraction protein PilT | transport | [102] |
| 85 | WP_003471666.1 | cytoplasmic | pilus retraction protein PilT | transport | [102] |
| 86 | WP_003467936.1 | cytoplasmic | 'cold-shock' DNA-binding domain | transcription regulation | [103] |
| 87 | WP_003472086.1 | cytoplasmic | 'cold-shock' DNA-binding domain | transcription regulation | [103] |
| 88 | WP_003488188.1 | cytoplasmic | 'cold-shock' DNA-binding domain | transcription regulation | [103] |
| 89 | WP_047324567.1 | cytoplasmic | histone deacetylase domain | transcription regulation | [104] |
| 90 | WP_003473181.1 | cytoplasmic | MerR HTH family regulatory protein | transcription regulation | [105] |
| 91 | WP_047324520.1 | cytoplasmic | Sigma-54 interaction domain | transcription regulation | [106] |
| 92 | WP_003468516.1 | cytoplasmic | OmpR/PhoB-type DNA-binding domain | transcription regulation | [107] |
| 93 | WP_003468538.1 | cytoplasmic | zinc finger, ClpX C4-type | transcription regulation | [108] |
| 94 | WP_003470669.1 | cytoplasmic | ribosomal protein L2, bacterial/organellar-type | protein biosynthesis | [109] |
| 95 | WP_003469157.1 | cytoplasmic | ribosomal protein S12/S23 | protein biosynthesis | [110] |
| 96 | WP_047324810.1 | cytoplasmic | tRNA synthetase class II core domain (G H P S and T) | protein biosynthesis | [111,112] |
| 97 | WP_047324986.1 | cytoplasmic | tRNA synthetases class II (A) | protein biosynthesis | [111,112] |
| 98 | WP_003472142.1 | cytoplasmic | tRNA synthetases class II (D K and N) | protein biosynthesis | [111,112] |
| 99 | WP_003472589.1 | cytoplasmic | tRNA synthetases class II (D K and N) | protein biosynthesis | [111,112] |
| 100 | WP_047324979.1 | cytoplasmic | tRNA synthetases class II (D K and N) | protein biosynthesis | [111,112] |

(*Continued.*)

**Table 1.** (Continued.)

| s. no. | sequence ID of putative Zn-binding protein | subcellular localization | functional domain/family | broad functional class | reference |
|--------|---------|---------|---------|---------|---------|
| 101 | WP_003466231.1 | periplasmic | CopA/multicopper oxidase | response to oxidative stress | [113] |
| 102 | WP_003470527.1 | periplasmic | copper/zinc superoxide dismutase (SOD) | response to oxidative stress | [114] |
| 103 | WP_038238706.1 | periplasmic | copper/zinc superoxide dismutase (SOD) | response to oxidative stress | [114] |
| 104 | WP_047325709.1 | cytoplasmic | maltose/galactoside acetyltransferase | response to oxidative stress | [115] |
| 105 | WP_003472119.1 | cytoplasmic | MutL C-terminal dimerization domain | DNA repair | [116] |
| 106 | WP_003467943.1 | cytoplasmic | TatD-related DNase | DNA repair | [117] |
| 107 | WP_047324617.1 | cytoplasmic | UvrA | DNA repair | [118] |
| 108 | WP_003465625.1 | outer-membrane | penicillin-binding protein 1B | antimicrobial resistance | [119,120] |
| 109 | WP_047324608.1 | inner-membrane | penicillin-binding protein 1C | antimicrobial resistance | [119,120] |
| 110 | WP_003467386.1 | cytoplasmic | RNase_PH | RNA processing | [121,122] |
| 111 | WP_080964784.1 | cytoplasmic | RNB domain | RNA processing | [121,122] |
| 112 | WP_004425670.1 | cytoplasmic | Clp ATPase | proteolysis | [123] |
| 113 | WP_003471337.1 | inner-membrane | peptidase family M41 | proteolysis | [124] |
| 114 | WP_003477578.1 | cytoplasmic | integrase core domain | DNA integration | [125] |
| 115 | WP_047324635.1 | cytoplasmic | integrase core domain | DNA integration | [125] |

It was documented earlier that Zn-binding proteins act as a cofactor for all the six types of enzymes, and therefore contribute significantly in various metabolic and other cellular processes, i.e. DNA repair, cell signalling, ROS detoxification and antimicrobial resistance [13,28,128–132]. Zinc is also known to bind proteins which are involved in gene expression and regulation, like sigma factor interacting proteins, RNA polymerases, tRNA synthetases, some ribosomal proteins and transcriptional factors [133–137]. In our study, SDR, response regulator receiver, ADH/GroES, tRNA synthetase and ABC transporter domains were commonly found in selected Zn-binding proteins. Further, the broad classification of these proteins based on domain description indicates their primary roles in metabolism (carbohydrates, proteins, lipid, nucleotides, etc.), proteolysis, protein biosynthesis, transport, cell signalling, transcription regulation protein folding and response to oxidative stress. The GO biological process network was in accordance with the domain-based broad classification. This network was also found to be enriched with the process of metabolism, signal transduction, proteolysis, lipid modification, response to ROS, protein peptide propyl isomerization and cellular ion homeostasis. Further, the GO molecular function network supports the findings of functional domain and GO biological process network, which signifies that most of the proteins involved in the metabolism have molecular activities of transition metal binding, nucleotide binding and NAD binding [9]. Also predicted Zn-binding proteins having hydrolase and transaminase activities may

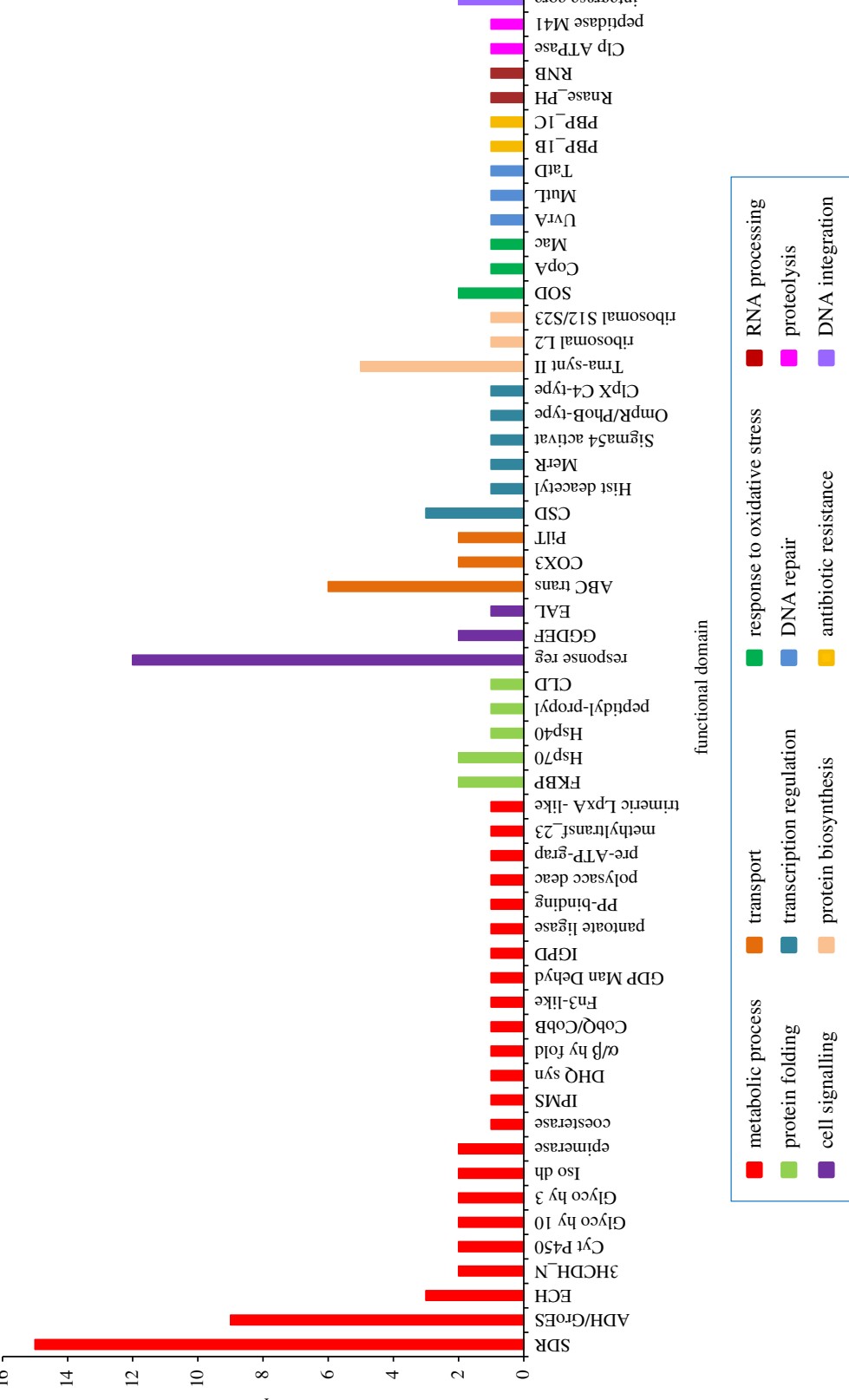

**Figure 6.** Functional domains of Zn-binding proteins probably involved in pathogen–host interaction. The graph here represents the functional domains of the Zn-binding proteins which are probably involved in pathogen–host interaction. The X-axis indicates functional domains and Y-axis shows number of proteins. SDR, response regulator, ADH/GroES, tRNA synthetase and ABC transporter were most common domains in these proteins. Colour of each bar indicate its broad functional class.

contribute in metabolic processes [13]. The proteins having cytochrome oxidase, peptidyl-propyl cis-trans isomerase and metallopeptidase activity probably specify their roles in transport, protein folding and proteolysis, respectively [91,100,138]. The subcellular localization of the proteins also determines their biological function [139]. The cellular compartmentalization of the Zn-binding proteins satisfies the fact that most of the proteins localized in cytoplasm are involved in metabolism, proteolysis, protein biosynthesis and cell signalling. Majority of the transporter proteins reside in inner-membrane.

Further, it was well documented earlier that Zn-binding proteins have considerable roles in bacterial toxin synthesis, virulence, antimicrobial resistance and host–pathogen interactions [14,140,141]. Here, we have found 115 PHI-base homologous Zn-binding proteins that probably engage in virulence and survival of *Xtu*. The Zn ions co-regulate the functioning of secretory proteins and contribute in plant–pathogen interactions [10], so secretory Zn-binding proteins have also been identified and checked for their putative roles in bacterial pathogenicity. Eleven proteins were found to be putative secretory Zn-binding proteins which were homologous to PHI-base.

These 115 proteins were categorized into 12 broad classes on the bases of domain description and their putative role in virulence (figure 6 and table 1). A total of 53 proteins were found in the category of metabolic process, out of which 15 proteins have SDR and nine proteins ADH/GroES domains. Earlier it was documented that SDR acts as a scaffold for redox sensor system and controls metabolic routes, transcription, cell signalling and stress, which further contributes in bacterial adaptation and pathogenesis [81,82]. ADH requires Zn metal ion for its catalytic activity and plays significant roles in alcohol fermentation, stress tolerance and virulence of bacteria [84–86]. Five secretory Zn-binding proteins were found in the class of metabolic process having glycoside hydrolase (4) and carboxylesterase (1) domain. It was stated previously that some glycoside hydrolases play role in host–microbe interactions and the enzymatic activity of some glycoside hydrolases was inhibited by $Zn^{2+}$ metal ion [70–72]. The carboxylesterase protein also has affinity for Zn ion [87] and its role to hydrolyse ester bonds and virulence of bacteria was also known earlier [88,142].

We have found seven proteins that belong to the category of protein folding; among these, five proteins were confined to cytoplasm having Hsp70/DnaK (2), Hsp40/DnaJ peptide (1), PPIase (1) and cyclophilin-type PPIase (1) domains. Two proteins were found putative secretory, out of seven. These two proteins are localized in periplasm and have FKBP-type PPIase domain. Earlier, Linke *et al.* [94] reported that Zn-centre II of DnaJ mediates the interactions between DnaJ and DnaK, which is crucial for closing the DnaK substrate binding site and for locking-in the substrate [94]. It was stated previously that PPIase activity of cyclophilin is inhibited by $Zn^{2+}$ ion in mouse macrophages cell line [89]. Further, a crystal study on Mip protein which is a propylisomerase of *Legionella pneumonia* indicated that $Zn^{2+}$ is required to mediate crystal contacts between the C-terminal FKBP domains of adjacent Mip dimmers [92]. Previous studies showed that all these domains are involved in catalysing the step of protein folding which helps the pathogens with stress adaptation, survival in harsh conditions and further aid in virulence [90,91,93].

A total of 15 cytoplasmic proteins were found in the category of cell signalling. Among these, 12 proteins have response regulator domain, two proteins have GGDEF and one protein has EAL domain. It was reported earlier that response regulator domain of two-component system is involved in cell-to-cell communication and adaptation to the different environment inside and outside host which is prerequisite for pathogenicity [97,98]. The role of GGDEF and EAL domains to mediate virulence of *Xanthomonas* has also been noted earlier [95,96].

Six inner-membrane ABC transporter proteins were identified in this study. Previously, it was known that ABC transporters are involved in import and export of Zn ions and help to maintain Zn homeostasis which contributes in bacterial virulence [18,99]. Two inner-membrane cytochrome C oxidase (COX3) proteins and two cytoplasmic pilus retraction proteins (PilT) have also been found in the category of transport. It was documented earlier that COX3 are present only in bacteria mainly in pathogenic bacteria and are critical for many anaerobic biological processes, colonizing low oxygen tissues, and biogenesis of oligomeric membrane proteins [101]. Also, the role of PilT proteins in twitching motility, cell adherence, biofilm formation and host colonization was known previously [102].

In the class of transcription regulation, we have found eight cytoplasmic Zn-binding proteins. Out of these, three proteins have Cold-shock DNA-binding domain (CSD). CSD containing proteins are evolutionarily conserved and extensively distributed nucleic acid binding proteins that aid in transcription regulation and are involved in numerous cellular processes like low-temperature adaptation, nutrients stress and cell growth. CSD proteins of plants have additional glycine-rich regions with CCHC-type zinc fingers. Kim *et al.* [143] reported that CSD proteins and glycine-rich RNA-binding proteins from *A. thaliana* help *E. coli* to grow and survive better in cold-shock condition, i.e. promote cold adaptation process [143]. The other five proteins in the category of transcription

regulation have histone deacetylase, Mer_HTH, Sigma-54 interaction, OmpR/PhoB-type DNA-binding and ClpX C4-type domains. All these domains require Zn for their catalytic activities, and also the roles of these in the regulation of transcription, virulence and pathogenicity of bacteria were previously documented in various studies [104–108].

Seven cytoplasmic proteins were found in the category of protein biosynthesis. Five of having tRNA synthetase II, one has ribosomal L2 and one has ribosomal S12/S23 domain. It was previously found that Zn ion helps in structural stability of these identified domains, which are involved in protein biosynthesis and also act as targets for many biocontrol agents [109–111,144]. A total of four proteins were found in the class of response to oxidative stress. Out of these, three were periplasmic secretory Zn-binding proteins having domains copper/zinc superoxide dismutase domain (SOD) (2) and CopA (1). The roles of these domains in radical oxygen species (ROS) detoxification, Zn homeostasis and bacterial virulence and survival were listed in earlier studies [113,114]. A cytoplasmic galactoside acetyltransferase domain containing protein was also found in this class, which is known to aid in cellular detoxification by acetylating non-metabolizable pyranosides [115].

We have categorized three cytoplasmic proteins having domains UvrA, MutL and TatD in the category of DNA repair. Earlier, it was reported that Zn ion is involved in structure architecture of Zn-finger domain of UvrA protein, and the role of C-terminal Zn-finger domain of UvrA protein has been noticed in regulation of damage-specific DNA binding [118]. MutL domain at its C-terminal contains a Zn-binding loop, a binding site for clamp DnaN and an endonuclease active site, which are critical for mismatch repair [116]. The TatD DNase domain has 3′-5′ exonuclease activity which digests single-stranded DNA and contributes in $H_2O_2$-induced DNA repair [117].

A secretory outer-membrane penicillin-binding protein (PBP) 1B and an inner-membrane PBP_1C were categorized in the class of antimicrobial resistance. Earlier, it was documented that PBP proteins not only localized in inner-membrane but also in outer-membrane, require Zn ion for their structural stability and are involved in antimicrobial resistance [119,120]. Two cytoplasmic proteins having RNase_PH/S1/KH and RNB domain were grouped in the class of RNA processing. Previous studies showed that RNase/RNB ribonucleases are critically required for RNA degradation, RNA and protein quality control and stress response, which results in raising virulence of the pathogens [121,122].

A cytoplasmic Clp ATPase domain containing protein and an inner-membrane peptidase M41 protein were categorized in class of proteolysis. Prior studies provide evidence that Clp ATPase causes cell proteolysis and plays central roles in virulence, gene expression, stress response and antimicrobial resistance [123]. Also, the role of Zn metallopeptidase M41 was previously listed in proteolysis and virulence [124].

Two cytoplasmic integrase core domain containing proteins were listed in the category of DNA integration. Formerly, it was known that bacterial integrase mediates site-specific recombination between bacterial and host cell [125] and further aids in pathogenesis.

# 5. Conclusion

To conclude, this study represents the first inclusive *in silico* report on Zn-binding proteins of *Xtu*. The functional diversity of Zn-binding proteins of *Xtu* unveil the facts that these proteins are metabolically versatile and contribute in various cellular and biological processes. The overall study provides the putative Zn-binding proteins repository and symbolizes their probable roles in growth, development, survival, pathogenicity and defence activities of *Xtu*. The presented repository may serve as starting material for experimental analysis which further paves the way to get insight into their mechanistic role in plant–pathogen interactions. Furthermore, in future these Zn-binding proteins may act as targets for designing metal-based antimicrobial agents in order to improve overall crop yield.

Data accessibility. The whole proteome *Xanthomonas translucens* pv. *undulosa* was downloaded from: (ftp:// ftp.ncbi.nlm.nih.gov/genomes/all/GCF/001/021/935/GCF_001021935.1_ASM102193v1).

Authors' contributions. S.K.V. and A.S. conceived the study. A.S. carried out the data analysis. D.S. assisted in structural analysis. A.S. drafted the manuscript. A.S. and S.K.V. edited the manuscript. All authors reviewed the manuscript and gave final approval for publication.

Competing interests. The authors have no competing interests.

Funding. S.K.V. laboratory was funded by Department of Biotechnology, India. A.S. and D.S. are thankful to Indian Council of Medical Research, India for their Senior Research Fellowships.

Acknowledgements. Central University of Himachal Pradesh is acknowledged by authors for providing computational facilities.

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
