## [Reviewer comments · Royal Society Open Science]

Review History

RSOS-190369.R0 (Original submission)

Review form: Reviewer 1

Is the manuscript scientifically sound in its present form?

No

Are the interpretations and conclusions justified by the results?

No

Is the language acceptable?

No

Is it clear how to access all supporting data?

Yes

Do you have any ethical concerns with this paper?

No

Have you any concerns about statistical analyses in this paper?

No

Recommendation?

Major revision is needed (please make suggestions in comments)

Comments to the Author(s)

See attached file (Appendix A).

Review form: Reviewer 2

Is the manuscript scientifically sound in its present form?

No

Are the interpretations and conclusions justified by the results?

No

Is the language acceptable?

Yes

Is it clear how to access all supporting data?

No

Do you have any ethical concerns with this paper?

No

Have you any concerns about statistical analyses in this paper?

No

Recommendation?

Major revision is needed (please make suggestions in comments)

Comments to the Author(s)

The manuscript focuses on a broadly important issue, the proteome of a bacterial plant pathogen. The importance of enumerating the metalloproteins of these organisms is well-described in the introduction. The authors focus on Zn-proteins to “understand the Zn homeostatic mechanism” (p. 3 line 59-60). They argue for non-experimental pipelines to rapidly identify metalloproteins and have presented such a work-flow here. Ultimately the mechanism of Zn-homeostasis (eg, balancing Zn supply with demand), which is well-studied in several bacteria (E. coli, B. subtilis, Staphylococcus) is not really addressed.

The work-flow has a logic to it, but a number of questions arise from the analysis that result from a loose definition of Zn sites. Critically, the value of the information provided is limited by the potential for false positives. Additionally, it is not clear whether this approach has identified novel Zn-proteins from Xanthomonas that would be interesting targets for inhibitor design because of the potential for high selectivity.

First, the authors rely on Metal PDB to define minimal functional Zn sites (p4. Line 30-31.) in the proteins from their organism. However, the outcome of this analysis (presented in part in Figure 1) shows a moderate abundance of Arg residues in Zn-binding sites. This is at odds with the

Metal-PDB and arises from the exceptionally long distance cutoff used to define Zn-sites. The reference used to support this cutoff is quite dated (from 1990) and a more stringent cutoff is recommended to at least define the primary coordination environment. A second range of distances could be used to capture residues more commonly found in 2nd and 3rd sphere coordination environments. Articles by Dudev and Lim provide detailed coverage of metal site geometries in proteins and could be used to guide the establishment of these boundaries. A second issue is whether all of these Zn-sites in the Metal PDB are actually Zn-loaded proteins in the cell, or are merely Zn-substituted proteins and the motif to which they bind is not evolved to bind Zn in the cell, thus losing its predictive value and creating false positives (see also later comments regarding Table S2).

Similarly, there is annotation of proteins as Zn-binding in various databases (section 3.2) which may result from data with purified proteins to which Zn will bind more tightly than metals such as Fe, which could be the physiological cofactor. Again, some false positives might emerge in the analysis.

The latter section looking for possible secreted Zn-proteins that could be involved in host-pathogen interaction is of interest but there is no critical consideration of whether these proteins are secreted under conditions linked to host-interactions.

Figure 7 is hard to understand. The Zn-coordination environments depicted here are incomplete, often showing two protein ligands when the expected minimal coordination would be three protein ligands. Additionally, it is not clear that the structures represent what the panel claims. For example, 7e is labelled as the ClpX four Cys Zn-finger but a four Cys Zn-finger is not represented.

Other comments:

Table 1 – Items 7 and 8 suggest a cytochrome P450 as a Zn-binding protein. These usually only bind heme so what feature makes these proteins link to Zn?

Supplementary Table 2 is very hard to navigate and thus use effectively. The binding site column shows too few residues in some cases (the first entry) and the Zn-ligand distances (final column) are sometimes not listed, are too long to be primary coordination interactions, or include an indication that the ligand is not one given in the binding site column. As indicated above in the comment for Figure 1, a rigorous cutoff defined by verified Zn-binding sites should be employed here. Also, as mentioned above, this table has captured Zn-substituted proteins that use a different metal for biological activity (eg, entry 270 cites PDBID 1irn, which is a Zn-substituted rubredoxin, normally an Fe protein).

The predicted protein structures using Phyre2 should be made available to allows others to assess the nature of the Zn site to enable study of these proteins (eg to generate site-directed mutants).

Decision letter (RSOS-190369.R0)

09-Jul-2019

Dear Dr Verma,

The editors assigned to your paper ("Zinc binding proteome of a phytopathogen *Xanthomonas translucens* pv. *undulosa*") have now received comments from reviewers.

Both reviewers raise a very significant number criticisms and concerns with the work and analysis as currently presented. These will all require very careful consideration. We would therefore like you to revise your paper in accordance with the referee's points which can be found below (not including confidential reports to the Editor). Please note this decision does not guarantee eventual acceptance.

Please submit a copy of your revised paper before 01-Aug-2019. Please note that the revision deadline will expire at 00.00am on this date. If we do not hear from you within this time then it will be assumed that the paper has been withdrawn. In exceptional circumstances, extensions may be possible if agreed with the Editorial Office in advance. We do not allow multiple rounds of revision so we urge you to make every effort to fully address all of the comments at this stage. If deemed necessary by the Editors, your manuscript will be sent back to one or more of the original reviewers for assessment. If the original reviewers are not available, we may invite new reviewers.

- Data accessibility

If you wish to submit your supporting data or code to Dryad (<http://datadryad.org/>), or modify your current submission to dryad, please use the following link:
<http://datadryad.org/submit?journalID=RSOS&manu=RSOS-190369>

- Competing interests

- Authors' contributions

- Acknowledgements

- Funding statement

on behalf of Dr Berat Haznedaroglu (Associate Editor) and Steve Brown (Subject Editor)
openscience@royalsociety.org

Comments to Author:

Reviewers' Comments to Author:

Reviewer: 1

Comments to the Author(s)

See attached file

Reviewer: 2

Comments to the Author(s)

The manuscript focuses on a broadly important issue, the proteome of a bacterial plant pathogen. The importance of enumerating the metalloproteins of these organisms is well-described in the introduction. The authors focus on Zn-proteins to “understand the Zn homeostatic mechanism” (p. 3 line 59-60). They argue for non-experimental pipelines to rapidly identify metalloproteins and have presented such a work-flow here. Ultimately the mechanism of Zn-homeostasis (eg, balancing Zn supply with demand), which is well-studied in several bacteria (*E. coli*, *B. subtilis*, *Staphylococcus*) is not really addressed.

The work-flow has a logic to it, but a number of questions arise from the analysis that result from a loose definition of Zn sites. Critically, the value of the information provided is limited by the potential for false positives. Additionally, it is not clear whether this approach has identified novel Zn-proteins from *Xanthomonas* that would be interesting targets for inhibitor design because of the potential for high selectivity.

First, the authors rely on Metal PDB to define minimal functional Zn sites (p4. Line 30-31.) in the proteins from their organism. However, the outcome of this analysis (presented in part in Figure 1) shows a moderate abundance of Arg residues in Zn-binding sites. This is at odds with the Metal-PDB and arises from the exceptionally long distance cutoff used to define Zn-sites. The reference used to support this cutoff is quite dated (from 1990) and a more stringent cutoff is recommended to at least define the primary coordination environment. A second range of distances could be used to capture residues more commonly found in 2nd and 3rd sphere coordination environments. Articles by Dudev and Lim provide detailed coverage of metal site geometries in proteins and could be used to guide the establishment of these boundaries. A second issue is whether all of these Zn-sites in the Metal PDB are actually Zn-loaded proteins in the cell, or are merely Zn-substituted proteins and the motif to which they bind is not evolved to bind Zn in the cell, thus losing its predictive value and creating false positives (see also later comments regarding Table S2).

Similarly, there is annotation of proteins as Zn-binding in various databases (section 3.2) which may result from data with purified proteins to which Zn will bind more tightly than metals such as Fe, which could be the physiological cofactor. Again, some false positives might emerge in the analysis.

The latter section looking for possible secreted Zn-proteins that could be involved in host-pathogen interaction is of interest but there is no critical consideration of whether these proteins are secreted under conditions linked to host-interactions.

Figure 7 is hard to understand. The Zn-coordination environments depicted here are incomplete, often showing two protein ligands when the expected minimal coordination would be three protein ligands. Additionally, it is not clear that the structures represent what the panel claims. For example, 7e is labelled as the ClpX four Cys Zn-finger but a four Cys Zn-finger is not represented.

Other comments:

Table 1 - Items 7 and 8 suggest a cytochrome P450 as a Zn-binding protein. These usually only bind heme so what feature makes these proteins link to Zn?

Supplementary Table 2 is very hard to navigate and thus use effectively. The binding site column shows too few residues in some cases (the first entry) and the Zn-ligand distances (final column)

are sometimes not listed, are too long to be primary coordination interactions, or include an indication that the ligand is not one given in the binding site column. As indicated above in the comment for Figure 1, a rigorous cutoff defined by verified Zn-binding sites should be employed here. Also, as mentioned above, this table has captured Zn-substituted proteins that use a different metal for biological activity (eg, entry 270 cites PDBID 1irn, which is a Zn-substituted rubredoxin, normally an Fe protein).

The predicted protein structures using Phyre2 should be made available to allow others to assess the nature of the Zn site to enable study of these proteins (eg to generate site-directed mutants).

Editorial Office Comments to Author:

For information about language editing services endorsed by the Royal Society, please follow the link below:

<https://royalsociety.org/journals/authors/language-polishing/>

Author's Response to Decision Letter for (RSOS-190369.R0)

See Appendix B.

RSOS-190369.R1 (Revision)

Review form: Reviewer 2

Is the manuscript scientifically sound in its present form?

Yes

Are the interpretations and conclusions justified by the results?

Yes

Is the language acceptable?

Yes

Do you have any ethical concerns with this paper?

No

Have you any concerns about statistical analyses in this paper?

No

Recommendation?

Accept as is

Comments to the Author(s)

The revised manuscript addresses concerns raised in review. The authors have done a nice job with the revision. The inclusion of the Phyre predicted structures (in the .rar file) means that a reader

can see the predicted site in combination with information in the Tables within the paper. This is particularly important for a researcher who, for instance, carries out a genetic screen and fits a hit in a gene that encodes a putative Zn-protein or enzyme. This study would enable that individual to make site-directed mutation to test the role of Zn in function.

Decision letter (RSOS-190369.R1)

21-Aug-2019

Dear Dr Verma,

I am pleased to inform you that your manuscript entitled "Zinc binding proteome of a phytopathogen *Xanthomonas translucens* pv. *undulosa*" is now accepted for publication in Royal Society Open Science.

on behalf of Dr Berat Haznedaroglu (Associate Editor) and Steve Brown (Subject Editor)
openscience@royalsociety.org

Reviewer comments to Author:
Reviewer: 2

Comments to the Author(s)
The revised manuscript address concerns raised in review. The authors have done a nice job with the revision. The inclusion of the Phyre predicted structures (in the .rar file) means that a reader

can see the predicted site in combination with information in the Tables within the paper. This is particularly important for a researcher who, for instance, carries out a genetic screen and finds a hit in a gene that encodes a putative Zn-protein or enzyme. This study would enable that individual to make site-directed mutation to test the role of Zn in function.

Appendix A

Review of Zinc-binding Proteome of a Phytopathogen *Xanthomonas*...

The authors present an in silico bioinformatics analysis of the genome of Xanthomonas translucens pv. Undulosa directed at uncovering potential Zn-binding proteins in the proteome and describing their functionality, particularly in relation to pathogenicity. The reviewer is an experimentalist working in the area of Zn-proteomics and Zn-trafficking. Thus, I will not directly assess the adequacy of the bioinformatics methods applied in this study. I will address other aspects of the manuscript.

First, the manuscript needs thorough editing to reach the standard of grammatical and idiomatic English (e.g. Page 1, line 50; Page 2, lines 18-23 and many more).

Second, I am sympathetic to this type of bioinformatics analysis which leads to the definition of a hypothetical Zn-proteome for this organism. But I am concerned that it is missing a lot of information that will allow the reader to assess the meaning and quality of the results. Therefore, I will move through the manuscript sequentially to raise concerns that the authors need to address.

Page 2, line 2 – 2:2 “This study provides primarily information to understand the Zn homeostatic mechanism of *Xtu*.”

2:60 “To understand the Zn homeostatic mechanism of a bacterial pathogen the key requirement is to accomplish inclusive facts of the Zn-binding proteins of that pathogen [19].”

3:17-18 “This primarily study helps in understanding the Zn homeostatic mechanism of *Xtu* which further may aids in controlling BLS.”

Will the results provide insight into Zn homeostatic mechanisms? I do not think so. Zn homeostasis maintains the correct Zn-proteome in the face of changing circumstances for the cell. But knowledge of the total Zn-proteome – the subject of this manuscript (3:11-13) - will not lead to an understanding of Zn homeostasis. So these statements need to be removed.

3:33-34 “ The proteins which found homologous to the Zn dataset of MetalPDB at expect value (e-value) ≤ 0.00001 were further selected.”

This statement is unclear. I think the Methods section 3.3 needs additional elaboration so that it is transparent to the non-expert exactly how the authors came up with their list of putative Zn-proteins.

3:48-54 “The effective radii of interaction for metal ion may vary from 3.5Å to 7Å [30]. In order to avoid the false positive predictions we have set the threshold for radii of interactions up to 5Å. The proteins which were found to interact with Zn ion up to 5Å were finally selected as putative Zn-binding proteins.”

Reference 30 is not focused on bond lengths (inner shell interactions) but the outer shell hydrophobic environment surrounding metal binding sites. Hardly any of the examples involve Zn. For a better reference, try See, RF, Kruse, RA, Strube, WM (1998) Inorg. Chem. 37: 5369-5375. Basically, metal-ligand bond lengths are in the 2-3 angstrom range not 3.5-5. So this needs to be corrected. (More about this later)

4:7 The word ‘edge’ is confusing. What is being described are ‘connections’ in GO diagrams. Please refer to connections or linkages.

4:49 *Table S1 and following tables. The column headings need to be described in full detail so that the reader can understand what the contents of the tables include. At present, column headings are in short hand and leave out needed detail.*

4:51-53 *“279 proteins showed interactions up to 5Å which were selected further as putative Zn-binding proteins (Electronic supplementary material, table S2).”*

Referring to Table S2, I looked at the proposed ligands for the first 25 structures. Only 4 had 3 or more. Generally, biological sites are either tri- or tetra-coordinate. The first concern about the list in S2 is that most (21/25) have only two ligands and will not bind tightly to Zn. If a site is designed to stabilize 3-D conformation, it is even less likely to bind Zn in stable fashion with only 2 ligands. And 5 of the 21 contain only carboxylate ligands, weaker binders than histidines and cysteines. Furthermore, a number of sites (7/25) have ligands that do not bind Zn to a significant extent, e.g. R, A, M. And, the bond lengths are very long, perhaps because the authors are modeling the structures based only on amino acid sequence in Phyre2. All of these concerns should be addressed in the text. It is particularly unlikely that Zn-containing structures with only two ligands will exist in vivo. Perhaps, they participate as part of the Zn buffering system (see reference 17, for example).

4:53 *What is meant by “the order of interactions”? From Figure 1 it seems that the intent is simply to show quantitatively how often different ligands are used in putative Zn-binding sites.*

5:24-25 *I am not sure that the GO figures 4 and 5 contribute much as presently formulated. Yes, there are lots of connections. But it is almost impossible to identify what specific GO biological processes (Table S4) are connected to one another because the writing is so tiny. These figures need to be revised.*

5:30-31 *“In order to check the regulation of biological process by Zn binding Proteins, the analysis of the network was made and number of interactions was estimated.” Define regulation and then explain how GO networks relate to regulation by Zn binding proteins. I would say that the GO networks show the ‘involvement’ of Zn binding proteins in metabolic processes not the ‘regulation’.*

6:4 *The Discussion seeks to relate the bioinformatics results with experimental results, showing that at least some of the proteins identified have already been shown to be Zn-proteins. That is important. Some of the references supporting the text that follows clearly relate members of the list of putative Zn-proteins to such experimental results. But a sampling of other references was unconvincing in providing this support. I will mention these as they appear in the manuscript.*

6:6 *“The present study focuses on the screening of Zn-binding proteins from the whole proteome of Xtu.”*

Please substitute, “The present study focuses on the bioinformatic identification of potential Zn-proteins” within the whole proteome of Xtu

6:6-18 *First paragraph of Discussion provides the only reflection on the results from the point of view of the bioinorganic chemistry of Zn-proteins. It is minimal and inadequate.*

What are the characteristics of the hypothetical Zn-proteome besides the list of ligands in supplementary Table S2 (note some ligands in supple. table are simply not Zn-binding ligands - K, A, M)? Lots of sites are two coordinate. At best these might be adventitious sites, which are involved in trafficking or buffering of Zn.

What if anything is known about Zn metabolism in this organism or related microorganisms?

It seems to me that a critical discussion of your findings as above is extremely important to assess the usefulness of the in silico approach as a starting point for an experimental understanding of the Zn-proteome in this microorganism.

6:20-21 “It is documented earlier that Zn-binding proteins contribute significantly in metabolism, transport, gene expression and regulation of other numerous cellular processes [14,19].”

Ref 14 focuses on Zn transport in experimental systems. Ref. 19 is an in silico study. There are many other references that focus on experimental results that might be cited here in support of the broad statement about Zn-protein function.

6:35-41 “Further, the GO molecular function network support the findings of functional domain and GO biological process network which signify that most of the proteins involved in the metabolism have molecular activities of transition metal binding, nucleotide binding, NAD binding, hydrolase and transaminase [9].”

Here is an example of a much too long sentence in need of editing for grammar and understanding.

Moreover, how does the GO network analysis demonstrate transition metal binding. Zn binding was the basis for the selection of proteins in this study in the first place.

6:40-44 “The proteins having cytochrome oxidase, peptidyl-propyl cis-trans isomerase and metallopeptidase activity probably specify their roles in transport,...”

Cytochrome oxidase is not usually classified functionally as a transporter, but as the terminal oxidase in electron transport from NADH.

6:51-52 “This analysis provides the clear evidence that Zn-binding proteins are critically required for growth and physiology of *Xtu*.”

This is too broad a statement. It is meaningless as all proteins are involved in physiology and many in growth. Moreover, as yet in the text the authors have not provided convincing evidence that these proteins are Zn-proteins, only that they are recognized as such in their bioinformatics analysis.

7:19-20 “Glycoside hydrolase require Zn metal for its structural stability and hydrolyze the plant cell wall which helps the pathogens to invade the host [61,62].”

Reference 61 does not discuss the Zn requirement and 62 is about Ca not Zn binding to the enzyme. These are incorrect references.

7:32-35 “Previous studies showed that all these domains have Zn-binding capability and involved in catalyzing the step of protein folding which helps the pathogens to survive in harsh conditions and further aid in virulence [54,66–68].”

Reference 52, for example, has only a fleeting reference to Zn binding and really does not support the authors' contention. I did not look at references 66 or 68.

8:2-5 “CSD containing proteins bind to Zn and DNA, aid in transcription regulation and are involved in numerous cellular processes like low temperature adaptation, nutrients stress and cell growth [76].”

In ref 76, the authors show 10 CSD structures from different microorganisms. Only one has a putative Zn binding site in an unusual C-terminal domain. This is another weak reference.

8:33-34 “UvrA is a Zn finger domain and it helps in regulation of damage-specific DNA binding [89].”

Regulation is the wrong word. Zn is simply a structural determinant of the protein.

An important contribution this paper can make is to look critically at the output of the in silico work flow. As far as I can tell, maybe 15% of the putative Zn-proteins have precedents in the literature as presented by the authors. The rest are truly hypothetical. But many of these with only 2 metal binding ligands are almost certainly not Zn-metalloproteins in vivo because they would not have sufficient thermodynamic stability. How the bioinformatics work flow decided that these were legitimate Zn-proteins despite their unfavorable stability should be discussed. Presumably, the database of metal sites in ref. 26 provided the set of metal binding motifs that was used. But if it included two carboxylate ligands as a motif for the binding of Zn, that is surely suspect.

Appendix B

Point by point response to the Reviewer 1 comments

Comment 1: *Review of Zinc-binding Proteome of a Phytopathogen Xanthomonas...*

The authors present an in silico bioinformatics analysis of the genome of Xanthomonas translucens pv. Undulosa directed at uncovering potential Zn-binding proteins in the proteome and describing their functionality, particularly in relation to pathogenicity. The reviewer is an experimentalist working in the area of Zn-proteomics and Zn-trafficking. Thus, I will not directly assess the adequacy of the bioinformatics methods applied in this study. I will address other aspects of the manuscript.

Our response 1: We are highly grateful to the learned reviewer for providing highly valuable suggestions to our manuscript.

Comment 2: *First, the manuscript needs thorough editing to reach the standard of grammatical and idiomatic English (e.g. Page 1, line 50; Page 2, lines 18-23 and many more).*

Second, I am sympathetic to this type of bioinformatics analysis which leads to the definition of a hypothetical Zn-proteome for this organism. But I am concerned that it is missing a lot of information that will allow the reader to assess the meaning and quality of the results. Therefore, I will move through the manuscript sequentially to raise concerns that the authors need to address.

Our response 2: We are thankful to the reviewer for providing valuable suggestions which helps us a lot to improve our manuscript. We have tried our best to answer the raised concerns of the reviewer and edited the manuscript grammatically.

Please see Page no. 1 line no. 44-52; Page No. 2 line no. 19-27.

Comment 3: Page 2, line 2 – 2:2 “This study provides primarily information to understand the Zn homeostatic mechanism of *Xtu*.”

2:60 “To understand the Zn homeostatic mechanism of a bacterial pathogen the key requirement is to accomplish inclusive facts of the Zn-binding proteins of that pathogen [19].”

3:17-18 “This primarily study helps in understanding the Zn homeostatic mechanism of *Xtu* which further may aids in controlling BLS.”

Will the results provide insight into Zn homeostatic mechanisms? I do not think so. Zn homeostasis maintains the correct Zn-proteome in the face of changing circumstances for the cell. But knowledge of the total Zn-proteome – the subject of this manuscript (3:11-13) - will not lead to an understanding of Zn homeostasis. So these statements need to be removed.

Our response 3: We have removed these statements from the revised manuscript.

Comment 4: 3:33-34 “The proteins which found homologous to the Zn dataset of MetalPDB at expect value (e-value) ≤ 0.00001 were further selected.”

This statement is unclear. I think the Methods section 3.3 needs additional elaboration so that it is transparent to the non-expert exactly how the authors came up with their list of putative Zn-proteins.

Our response 4: We have now elaborated the Material and Method section of the revised manuscript. Please see Page no. 3 line no. 54-60; Page no. 4 line no. 1-45.

Comment 5: 3:48-54 “The effective radii of interaction for metal ion may vary from 3.5Å to 7Å [30]. In order to avoid the false positive predictions we have set the threshold for radii of interactions up to 5Å. The proteins which were found to interact with Zn ion up to 5Å were finally selected as putative Zn-binding proteins.”

Reference 30 is not focused on bond lengths (inner shell interactions) but the outer shell hydrophobic environment surrounding metal binding sites. Hardly any of the examples involve Zn. For a better reference, try See, RF, Kruse, RA, Strube, WM (1998) Inorg. Chem. 37: 5369-5375. Basically, metal-ligand bond lengths are in the 2-3 angstrom range not 3.5-5. So this needs to be corrected. (More about this later)

Our response 5: We have now improved the statement and provided correct references. Please see Page no. 4 line no. 31-44.

Comment 6: 4:7 The word ‘edge’ is confusing. What is being described are ‘connections’ in GO diagrams. Please refer to connections or linkages.

Our response 6: In the network terminology node and edge terms are used. Nodes are object and the connections between the objects are represented as edges. Therefore, we have used these terms.

We have now replaced the word edge with connections in the revised manuscript except at one point in Material and method section. Page no. 4 line no. 31-45.

Comment 7: 4:49 Table S1 and following tables. The column headings need to be described in full detail so that the reader can understand what the contents of the tables include. At present, column headings are in short hand and leave out needed detail.

Our response 7: We have modified the column headings in the revised Supplementary Tables. (Changes are highlighted in **yellow color**)

Comment 8: 4:51-53 “279 proteins showed interactions up to 5Å which were selected further as putative Zn-binding proteins (Electronic supplementary material, table S2).”

Referring to Table S2, I looked at the proposed ligands for the first 25 structures. Only 4 had 3 or more. Generally, biological sites are either tri- or tetra-coordinate. The first concern about the list in S2 is that most (21/25) have only two ligands and will not bind tightly to Zn. If a site is designed to stabilize 3-D conformation, it is even less likely to bind Zn in stable fashion with only 2 ligands. And 5 of the 21 contain only carboxylate ligands, weaker binders than histidines and cysteines. Furthermore, a number of sites (7/25) have ligands that do not bind Zn to a significant extent, e.g. R, A, M. And, the bond lengths are very long, perhaps because the authors are modeling the structures based only on amino acid sequence in Phyre2. All of these concerns should be addressed in the text. It is particularly unlikely that Zn-containing structures with only two ligands will exist in vivo. Perhaps, they participate as part of the Zn buffering system (see reference 17, for example).

Our response 8: We have used MetalPDB database for the collection of Zn-binding proteins which were already known and specified in Protein Data Bank. We have prepared local dataset of these known Zn-binding proteins and performed BlastP search of *Xtu* proteome with this database. The proteins which showed homology with this local database at e-value less than equal to 0.00001 were listed as putative Zn-binding proteins having Zn-binding sequence motifs. Further to determine structural Zn-binding sites in the shortlisted

proteins we have used Metal ion binding prediction and docking server (MIB). This server takes input file in PDB format, so we have first modeled the 3D-structures of the shortlisted proteins by Phyre2 server and then performed MIB search. MIB server is based on a fragment transformation method. In this method the query protein was aligned to the metal binding templates those were extracted from metal bound proteins present in PDB. The templates represent the local structure of metal binding residues within 3.5Å. According to MIB server a metal binding site had to contain a metal ion and at least two residues to quantify as a metal ion binding residue template. Each cluster after sequence and structural similarity acquire a score which is used for prediction of metal binding sites. For the evaluation of sequence similarity, MIB server utilizes BLOSUM62 matrix and for calculation of structural similarity root mean square deviation of C_α atoms of the alignments was used. At more than 95% specificity threshold, MIB server predicts the Zn-binding proteins with 94.8% accuracy and 71.1% sensitivity. To check the interactions of MIB docked metal ion with the protein we have used Ligplot⁺ visualization tool. We found that interacting residues and interaction radii provided by MIB may vary in wide range. Therefore, the interaction distance was raised from primary sphere (up to 3.5Å) (provided by MIB server) to secondary sphere (5Å). As it was stated earlier that second shell of interactions helps in stabilizing metal binding site, raise metal affinity and play role in determining physical properties of transition metal complexes (Shook and Borovik, 2010; Ngo et al., 2015; Dudev and Lim 2013).

- Shook, R. L., & Borovik, A. S. (2010). Role of the secondary coordination sphere in metal-mediated dioxygen activation. *Inorganic chemistry*, 49(8), 3646-3660.
- Ngo, V., da Silva, M. C., Kubillus, M., Li, H., Roux, B., Elstner, M., ... & Noskov, S. Y. (2015). Quantum effects in cation interactions with first and second coordination shell ligands in metalloproteins. *Journal of chemical theory and computation*, 11(10), 4992-5001.
- Dudev, T., & Lim, C. (2013). Competition among metal ions for protein binding sites: determinants of metal ion selectivity in proteins. *Chemical reviews*, 114(1), 538-556.

Earlier it was stated that in a cell the cytosolic Zn-binding proteins, transporters localized in cytoplasmic membranes and sensors of cytoplasmic free Zn ions are the molecules involved in Zn-homeostatic mechanisms (Colvin et al., 2010). In our study we have provided the primarily information of Zn-proteome of *Xtu*.

- Colvin RA, Holmes WR, Fontaine CP, Maret W. 2010 Cytosolic zinc buffering and muffling: Their role in intracellular zinc homeostasis. *Metallomics* 2, 306–317.

We have used MetalPDB and MIB servers which rely on PDB database as a source and contain not only Zn-binding proteins but also have Zn-substituted proteins and proteins participate as part of Zn buffering system. In PDB database biological assemblies of some proteins are available and some of the proteins are present as asymmetric unit, shows only one or two residues binding to the metal atom. In biological active form many of the proteins exists as more than one copy (dimer, trimer, tetramer and hexamer) and metal at their interface. Further, in some cases ligand atom of the protein may bind to metal ion for their stable coordination in biological active form.

Therefore, our *in-silico* prediction we have not determined the coordination geometry of the proteins, but we have found the proteins having Zn-binding motifs and found their probable binding site based on fragment transformation method.

In the revised manuscript we have now provided this information.

Please see Material and Method section 3.1 Page no. 3 line no. 44-60; Page no. 4 line no. 1-45; Discussion section Page no. 7 line no 11-30.

Comment 9: 4:53 *What is meant by “the order of interactions”? From Figure 1 it seems that the intent is simply to show quantitatively how often different ligands are used in putative Zn-binding sites.*

Our response 9: We have now corrected the statement in revised manuscript.

Comment 10: 5:24-25 *I am not sure that the GO figures 4 and 5 contribute much as presently formulated. Yes, there are lots of connections. But it is almost impossible to identify what specific GO biological processes (Table S4) are connected to one another because the writing is so tiny. These figures need to be revised.*

Our response: The Gene ontology represents the common classification scheme for gene function. The GO network was constructed based on the GO of the predicted Zn-binding proteins using ClueGO plug in cytoscape. Regarding the size of GO terms level in the figures we are unable to raise their size as this figure is generated by cytoscape. Although, we have now improved the Table S4.

In these tables the GO terms that belong to same groups are connected to each other. The GO term present in multiple groups are also connected to other groups which indicate their probable role in multiple processes. Grouping is based on Kappa score. Further we have also provided the list of GO term associated genes which belong to predicted Zn-binding proteins in supplementary tables (4 and 5). The connections between the GO terms and genes were not shown in the networks (figures) as it is hard to show all the connections provided in these complex networks.

Comment 11: 5:30-31 *“In order to check the regulation of biological process by Zn binding Proteins, the analysis of the network was made and number of interactions was estimated.” Define regulation and then explain how GO networks relate to regulation by Zn binding proteins. I would say that the GO networks show the ‘involvement’ of Zn binding proteins in metabolic processes not the ‘regulation’.*

Our response 11: Yes, we are agreed with the reviewer comment. We have now replaced the word regulation with involvement. Please see: Page no. 6 line no.22-23.

Comment 12: 6:4 *The Discussion seeks to relate the bioinformatics results with experimental results, showing that at least some of the proteins identified have already been shown to be Zn-proteins. That is important. Some of the references supporting the text that follows clearly relate members of the list of putative Zn-proteins to such experimental results. But a sampling of other references was unconvincing in providing this support. I will mention these as they appear in the manuscript.*

Our response 12: We are thankful to the reviewer for reviewing our manuscript critically and providing valuable suggestions. We have put our full efforts to improve the manuscript.

Comment 13: 6:6 *“The present study focuses on the screening of Zn-binding proteins from the whole proteome of Xtu.”*

Please substitute, “The present study focuses on the bioinformatic identification of potential Zn-proteins” within the whole proteome of Xtu

Our response 13: We have changed the statement in the revised manuscript.

Comment 14: 6:6-18 First paragraph of Discussion provides the only reflection on the results from the point of view of the bioinorganic chemistry of Zn-proteins. It is minimal and inadequate.

What are the characteristics of the hypothetical Zn-proteome besides the list of ligands in supplementary Table S2 (note some ligands in supple. table are simply not Zn-binding ligands - K, A, M)? Lots of sites are two coordinate. At best these might be adventitious sites, which are involved in trafficking or buffering of Zn.

What if anything is known about Zn metabolism in this organism or related microorganisms?

It seems to me that a critical discussion of your findings as above is extremely important to assess the usefulness of the in silico approach as a starting point for an experimental understanding of the Zn-proteome in this microorganism.

Our response: We are thankful to the reviewer for this suggestion. We are now modified the discussion. **Please see Page no. 7 line no.11-40.**

Comment 15: 6:20-21 “ It is documented earlier that Zn-binding proteins contribute significantly in metabolism, transport, gene expression and regulation of other numerous cellular processes [14,19].”

Ref 14 focuses on Zn transport in experimental systems. Ref. 19 is an in silico study. There are many other references that focus on experimental results that might be cited here in support of the broad statement about Zn-protein function.

Our response 15: We have now revised the statement and provided more supported references. **Please see Page no. 7 line no. 31-40.**

Comment 16: 6:35-41 “Further, the GO molecular function network support the findings of functional domain and GO biological process network which signify that most of the proteins involved in the metabolism have molecular activities of transition metal binding, nucleotide binding, NAD binding, hydrolase and transaminase [9].”

Here is an example of a much too long sentence in need of editing for grammar and understanding.

Moreover, how does the GO network analysis demonstrate transition metal binding. Zn binding was the basis for the selection of proteins in this study in the first place.

Our response 16: We have corrected the sentence. **Please see Page no. 7 line 57-60.**

The Gene ontology represents the common classification scheme for gene function (Ashburner et al., 2000) which allow unification of biology. Three categories of GO are: GO biological process, GO molecular activities and GO cellular components. GO biological process indicate the biological objective of a gene. GO molecular function indicate the biochemical activity (including specific binding to ligands or structures) of a gene product. GO cellular component indicate the subcellular localization of the gene.

The transition metal binding indicates here a GO group in our predicted Zn-binding proteins which contain GO molecular terms those are connected to each other based on shared genes.

- Ashburner, M., Ball, C. A., Blake, J. A., Botstein, D., Butler, H., Cherry, J. M., ... & Harris, M. A. (2000). Gene ontology: tool for the unification of biology. *Nature genetics*, 25(1), 25.

Comment 17: 6:40-44 “The proteins having cytochrome oxidase, peptidyl-propyl cis-trans isomerase and metallopeptidase activity probably specify their roles in transport...”

Cytochrome oxidase is not usually classified functionally as a transporter, but as the terminal oxidase in electron transport from NADH.

Our response 17: We are agreed with the reviewer's statement that Cytochrome oxidase (Cox) proteins are involved in oxygen reduction, but in broad prospects Cox is the last enzyme in the electron transport chain, catalyzing electron transport (reduction of oxygen to water). As it receives electron from cytochrome c molecules and transfers them to oxygen coupled to transmembrane proton gradient and converting it in to water. Therefore, we have broadly classified these proteins in the category of transport (Ludwig, 1987).

- Ludwig,B.(1987).Cytochrome c oxidase in prokaryotes. *FEMS Microbiol. Lett.* 46, 41–56. doi:10.1016/0378-1097(87)90185-6

Comment 18: 6:51-52 “This analysis provides the clear evidence that Zn-binding proteins are critically required for growth and physiology of *Xtu*.”

This is too broad a statement. It is meaningless as all proteins are involved in physiology and many in growth. Moreover, as yet in the text the authors have not provided convincing evidence that these proteins are Zn-proteins, only that they are recognized as such in their bioinformatics analysis.

Our response 18: We have removed the statement from the updated manuscript.

Comment 19: 7:19-20 “Glycoside hydrolase require Zn metal for its structural stability and hydrolyze the plant cell wall which helps the pathogens to invade the host [61,62].”

Reference 61 does not discuss the Zn requirement and 62 is about Ca not Zn binding to the enzyme. These are incorrect references.

Our response 19: We have now corrected the statement and provided correct references. **Please see Page no. 8 line no.39-45.**

Comment 20: 7:32-35 “Previous studies showed that all these domains have Zn-binding capability and involved in catalyzing the step of protein folding which helps the pathogens to survive in harsh conditions and further aid in virulence [54,66–68].”

Reference 52, for example, has only a fleeting reference to Zn binding and really does not support the authors' contention. I did not look at references 66 or 68.

Our response 20: We have now corrected the statement and provided correct references in the revised manuscript. Please see **Page No. 8 line no. 55-60; Page no. 9 line no. 1-7.**

Comment 21: 8:2-5 “CSD containing proteins bind to Zn and DNA, aid in transcription regulation and are involved in numerous cellular processes like low temperature adaptation, nutrients stress and cell growth [76].”

In ref 76, the authors show 10 CSD structures from different microorganisms. Only one has a putative Zn binding site in an unusual C-terminal domain. This is another weak reference.

Our response 21: We have now corrected the statement and provided new references. Please see Page no. 9 line no. 36-47.

Comment 22: 8:33-34 “UvrA is a Zn finger domain and it helps in regulation of damage-specific DNA binding [89].”

Regulation is the wrong word. Zn is simply a structural determinant of the protein.

Our response 22: Yes, we are agreed with the reviewer’s statement Zinc is the structural determinant of the protein. We have now corrected the line. Please see Page no. 10 line no. 10-15.

Comment 23: *An important contribution this paper can make is to look critically at the output of the in silico work flow. As far as I can tell, maybe 15% of the putative Zn-proteins have precedents in the literature as presented by the authors. The rest are truly hypothetical. But many of these with only 2 metal binding ligands are almost certainly not Zn-metalloproteins in vivo because they would not have sufficient thermodynamic stability. How the bioinformatics work flow decided that these were legitimate Zn-proteins despite their unfavorable stability should be discussed. Presumably, the database of metal sites in ref. 26 provided the set of metal binding motifs that was used. But if it included two carboxylate ligands as a motif for the binding of Zn, that is surely suspect.*

Our response 23: This study provides an *in-silico* report on Zn-binding proteins of Xtu. For the prediction of Zn-binding proteins and their functional annotation we rely on bioinformatics servers and the supported literature. The predicted proteins are putative and need further experimental validation. Regarding the metal binding site and stability of the proteins we have explained earlier in above comment that computational databases and tools not only rely on biological assemblies but also on asymmetric unit.

Further, regarding the workflow of the study we have not only predicted the Zn-binding proteins but also annotated these based on their domains and GO. Also, we have provided the supported literature which indicate that these predicted proteins probably have affinity for Zn.

Point by point response to the Reviewer 2 comments

Comment 1: Comments to the Author(s)

The manuscript focuses on a broadly important issue, the proteome of a bacterial plant pathogen. The importance of enumerating the metalloproteins of these organisms is well-described in the introduction. The authors focus on Zn-proteins to “understand the Zn homeostatic mechanism” (p. 3 line 59-60). They argue for non-experimental pipelines to rapidly identify metalloproteins and have presented such a work-flow here. Ultimately the mechanism of Zn-homeostasis (eg, balancing Zn supply with demand), which is well-studied in several bacteria (*E. coli*, *B. subtilis*, *Staphylococcus*) is not really addressed.

Our response1: We are highly thankful to the reviewer for reviewing our manuscript critically.

We have now addressed the Zn homeostasis mechanism of some bacteria. Please see Page no. 3 line no. 1-16.

Comment 2: The work-flow has a logic to it, but a number of questions arise from the analysis that result from a loose definition of Zn sites. Critically, the value of the information provided is limited by the potential for false positives. Additionally, it is not clear whether this approach has identified novel Zn-proteins from *Xanthomonas* that would be interesting targets for inhibitor design because of the potential for high selectivity.

First, the authors rely on Metal PDB to define minimal functional Zn sites (p4. Line 30-31.) in the proteins from their organism. However, the outcome of this analysis (presented in part in Figure 1) shows a moderate abundance of Arg residues in Zn-binding sites. This is at odds with the Metal-PDB and arises from the exceptionally long distance cutoff used to define Zn-sites. The reference used to support this cutoff is quite dated (from 1990) and a more stringent cutoff is recommended to at least define the primary coordination environment. A second range of distances could be used to capture residues more commonly found in 2nd and 3rd sphere coordination environments. Articles by Dudev and Lim provide detailed coverage of metal site geometries in proteins and could be used to guide the establishment of these boundaries. A second issue is whether all of these Zn-sites in the Metal PDB are actually Zn-loaded proteins in the cell, or are merely Zn-substituted proteins and the motif to which they bind is not evolved to bind Zn in the cell, thus losing its predictive value and creating false positives (see also later comments regarding Table S2).

Our response 2: We are highly thankful to the reviewer for reviewing our manuscript. We have used MetalPDB database for the collection of Zn-binding proteins which were already known and specified in Protein Data Bank. We have prepared local dataset of these known Zn-binding proteins and performed BlastP search of *Xtu* proteome with this database. The proteins which showed homology with this local database at e-value less than equal to 0.00001 were listed as putative Zn-binding proteins having Zn-binding sequence motifs. Further to determine structural Zn-binding sites in the shortlisted proteins we have used Metal ion binding prediction and docking server (MIB). This server takes input file in PDB format so we have first modeled the 3D-structures of the shortlisted proteins by Phyre2 server and then performed MIB search. MIB server based on a fragment transformation method. In this method the query protein was aligned to the metal binding templates those were extracted from metal bound proteins present in PDB. The templates represent the local structure of metal binding residues within 3.5Å. According to MIB server a metal binding site had to contain a metal ion and at least two residues to quantify as a metal ion

binding residue template. Each cluster after sequence and structural similarity acquire a particular score which is used for prediction of metal binding sites. For the evaluation of sequence similarity, MIB server utilizes BLOSUM62 matrix and for calculation of structural similarity root mean square deviation of C α atoms of the alignments was used. At more than 95% specificity threshold, MIB server predicts the Zn-binding proteins with 94.8% accuracy and 71.1% sensitivity. To check the interactions of MIB docked metal ion with the protein we have used Ligplot⁺ visualization tool. We found that interacting residues and interaction radii provided by MIB may vary in wide range. Therefore, the interaction distance was raised from primary sphere (up to 3.5Å) (provided by MIB server) to secondary sphere (5Å). As it was stated earlier that second shell of interactions helps in stabilizing metal binding site, raise metal affinity and play role in determining physical properties of transition metal complexes (Shook and Borovik, 2010; Ngo et al., 2015; Dudev and Lim 2013).

- Shook, R. L., & Borovik, A. S. (2010). Role of the secondary coordination sphere in metal-mediated dioxygen activation. *Inorganic chemistry*, 49(8), 3646-3660.
- Ngo, V., da Silva, M. C., Kubillus, M., Li, H., Roux, B., Elstner, M., ... & Noskov, S. Y. (2015). Quantum effects in cation interactions with first and second coordination shell ligands in metalloproteins. *Journal of chemical theory and computation*, 11(10), 4992-5001.
- Dudev, T., & Lim, C. (2013). Competition among metal ions for protein binding sites: determinants of metal ion selectivity in proteins. *Chemical reviews*, 114(1), 538-556.

Yes, MetalPDB database contain all Zn-binding sites i.e all Zn-loaded proteins in the cell or Zn-substituted proteins. Further, the logic behind their consideration is that the Zn substituted proteins may have potential to bind Zn ion also. Therefore, it may act as putative target to inhibit the activity of that protein or for designing metal-based inhibitors.

Please see Material and Method section 3.1 Page no. 3 line no. 44-60; Page no. 4 line no. 1-45; Discussion section Page no. 7 line no 11-30.

Comment 3: Similarly, there is annotation of proteins as Zn-binding in various databases (section 3.2) which may result from data with purified proteins to which Zn will bind more tightly than metals such as Fe, which could be the physiological cofactor. Again, some false positives might emerge in the analysis.

Our response 3: As we mentioned in the above response that MetalPDB contained all Zn-binding proteins including Zn substituted proteins. So, our prediction also includes putative Zn substituted proteins. Further, the logic behind their consideration is that the Zn substituted proteins may have potential to bind Zn ion also. Therefore, it may act as putative target to inhibit the activity of that particular protein or for designing metal-based inhibitors.

Comment 4: The latter section looking for possible secreted Zn-proteins that could be involved in host-pathogen interaction is of interest but there is no critical consideration of whether these proteins are secreted under conditions linked to host-interactions.

Our response 4: In this *in silico* study we have first checked that putative Zn binding proteins showed similarity with Pathogen Host interaction database (PHI-database). PHI-database

contains experimentally validated virulent and effector proteins which were known to play role in pathogen-host interactions. We have shortlisted the proteins which were homologous to PHI-database and considered that these may involve in plant-pathogen interactions. Further, from these we have extracted the secreted proteins. So, we have putatively considered that these secreted Zn-binding proteins probably involved in pathogen-host interactions.

Comment 5: Figure 7 is hard to understand. The Zn-coordination environments depicted here are incomplete, often showing two protein ligands when the expected minimal coordination would be three protein ligands. Additionally, it is not clear that the structures represent what the panel claims. For example, 7e is labelled as the ClpX four Cys Zn-finger but a four Cys Zn-finger is not represented.

Our response 5: In this study we have tried to find the proteins whether the protein is Zn-binding or not i.e proteins have ability to bind Zn^{2+} or not. For this prediction we rely on MetalPDB database and MIB server. As stated, above MetalPDB database contain all Zn-binding proteins including Zn-substituted proteins. And secondly, we have used MIB server which is based on fragment transformation method and uses a template for prediction having a metal ion and at least two interacting residues within 3.5Å. Further, both the server does not only rely on biological assemblies of the proteins as uses PDB database as a source. In biological active form many of the proteins exists as more than one copy (dimer, trimer, tetramer and hexamer) and metal at their interface. In PDB database some proteins are present as asymmetric unit shows only one or two residues binding to the metal atom. Further, in some cases ligand atom may bind to metal ion for their stable coordination in biological active form. Therefore, our *in-silico* prediction we have not determined the coordination geometry of the proteins, but we have found the proteins having Zn-binding motifs and found their probable binding site based on fragment transformation method. In figure 7, we have simply showed some representative structures of the putative Zn-binding proteins. Yes, we agreed with the reviewer that this figure do not contribute much in our study, so we have removed this figure in the revised manuscript. Secondly, figure 7e represents the interacting residues predicted by MIB server which is based on fragment transformation method. Further, when we analyzed the given protein for its domain using InterProScan and Pfam databases we found that it contains ClpX domain. This may be because MIB used fragment transformation method for prediction of binding site and docking and at more than 95% specificity threshold, MIB server predicts the Zn-binding sites with 94.8% accuracy and 71.1% sensitivity.

Other comments:

Comment 6: Table 1 – Items 7 and 8 suggest a cytochrome P450 as a Zn-binding protein. These usually only bind heme so what feature makes these proteins link to Zn?

Our response 6: Yes, we are agreed with the reviewer that cytochrome P450 is a heme binding protein. But it may have binding affinity for Zn^{2+} . Various studies are made on Zn-substituted cytochrome P450. As we mentioned in the above response that MetalPDB contained all Zn-binding proteins including Zn substituted proteins and hence our study also has all these proteins. Further, a study indicates that Zn^{2+} ion have inhibitory effect on human cytochrome P450 3A4 activity.

Please see the references

- Jankowska, K. I., Pagba, C. V., & Piotrowiak, P. (2012). Zinc-Substituted Cytochrome P450cam: Characterization of Protein Conformers F420 and F450 by Photoinduced Electron Transfer. *Biochemistry*, 51(7), 1431-1438.
- Kim, J. S., & Yun, C. H. (2005). Inhibition of human cytochrome P450 3A4 activity by zinc (II) ion. *Toxicology letters*, 156(3), 341-350.

Comment 7: Supplementary Table 2 is very hard to navigate and thus use effectively. The binding site column shows too few residues in some cases (the first entry) and the Zn-ligand distances (final column) are sometimes not listed, are too long to be primary coordination interactions, or include an indication that the ligand is not one given in the binding site column. As indicated above in the comment for Figure 1, a rigorous cutoff defined by verified Zn-binding sites should be employed here. Also, as mentioned above, this table has captured Zn-substituted proteins that use a different metal for biological activity (eg, entry 270 cites PDBID 1irn, which is a Zn-substituted rubredoxin, normally an Fe protein).

Our response 7: The binding site in first column is provided by MIB server within 3.5 Å.

Further, to check the interactions of MIB docked metal ion with the protein we have used Ligplot⁺ visualization tool. We found that interacting residues and interaction radii provided by MIB may vary in wide range. Therefore the interaction distance was raised from primary sphere (up to 3.5Å) (provided by MIB server) to secondary sphere (5Å). As it was stated earlier that second shell of interactions helps in stabilizing metal binding site, raise metal affinity and play role in determining physical properties of transition metal complexes (Shook and Borovik, 2010; Ngo et al., 2015; Dudev and Lim 2013).

- Shook, R. L., & Borovik, A. S. (2010). Role of the secondary coordination sphere in metal-mediated dioxygen activation. *Inorganic chemistry*, 49(8), 3646-3660.
- Ngo, V., da Silva, M. C., Kubillus, M., Li, H., Roux, B., Elstner, M., ... & Noskov, S. Y. (2015). Quantum effects in cation interactions with first and second coordination shell ligands in metalloproteins. *Journal of chemical theory and computation*, 11(10), 4992-5001.
- Dudev, T., & Lim, C. (2013). Competition among metal ions for protein binding sites: determinants of metal ion selectivity in proteins. *Chemical reviews*, 114(1), 538-556.

Therefore, last column indicates the binding residues within MIB docked proteins up to interacting distance 5Å. Ligplot⁺ is not able to measure covalent bond. Therefore, Zn-ligand distances are sometimes not listed in final column. Further, regarding Zn-substituted proteins we have mentioned in the above response that our data also have Zn-substituted proteins. We have now modified the column heading in supplementary table S2.

Comment 8: The predicted protein structures using Phyre2 should be made available to allows others to assess the nature of the Zn site to enable study of these proteins (e.g. to generate site-directed mutants).

Our response 8: In Supplementary table S1 we have listed all the information of the modeled structures provided by Phyre2. We can also provide structures modeled by Phyre2.